# DYNAMIC PARAMETERIZED NETWORK FOR CTR PREDICTION

## ABSTRACT

Learning to capture feature relations effectively and efficiently is essential in click-through rate (CTR) prediction of modern recommendation systems. Most existing CTR prediction methods model such relations either through tedious manually-designed low-order interactions or through inflexible and inefficient high-order interactions, which both require extra DNN modules for implicit interaction modeling. In this paper, we proposed a novel plug-in operation, *Dynamic Parameterized Operation* (DPO), to learn both explicit and implicit interaction instance-wisely. We showed that the introduction of DPO into DNN modules and Attention modules can respectively benefit two main tasks in CTR prediction, enhancing the adaptiveness of feature-based modeling and improving user behavior modeling with the instance-wise locality. Our *Dynamic Parameterized Networks* significantly outperforms state-of-the-art methods in the offline experiments on the public dataset and real-world production dataset, together with an online A/B test. Furthermore, the proposed *Dynamic Parameterized Networks* has been deployed in the ranking system of one of the world's largest e-commerce companies, serving the main traffic of hundreds of millions of active users.

## 1 INTRODUCTION

Click-through rate (CTR) prediction, which aims to estimate the probability of a user clicking an item, is of great importance in recommendation systems and online advertising systems (Cheng et al., 2016; Guo et al., 2017; Rendle, 2010; Zhou et al., 2018b). Effective feature modeling and user behavior modeling are two critical parts of CTR prediction.

Deep neural networks (DNNs) have achieved tremendous success on a variety of CTR prediction methods for feature modeling (Cheng et al., 2016; Guo et al., 2017; Wang et al., 2017). Under the hood, its core component is a linear transformation followed by a nonlinear function, which models weighted interaction between the flattened inputs and contexts by fixed kernels, regardless of the intrinsic decoupling relations from specific contexts (Rendle et al., 2020). This property makes DNN learn interaction in an implicit manner, while limiting its ability to model explicit relation, which is often captured by feature crossing component (Rendle, 2010; Song et al., 2019). Most existing solutions exploit a combinatorial framework (feature crossing component + DNN component) to leverage both implicit and explicit feature interactions, which is suboptimal and inefficient (Cheng et al., 2016; Wang et al., 2017). For instance, *wide & deep* combines a linear module in the wide part for explicit low-order interaction and a DNN module to learn high-order feature interactions. Follow-up works such as Deep & Cross Network (DCN) follows a similar manner by replacing the wide part with more sophistic networks, however, posits restriction to input size which is inflexible.

Above-mentioned methods pay little attention to user behavior modeling. Recently, attention-based methods like DIN and DIEN have attracted many interests that attempt to capture user preferences based on users' historical behaviors (Zhou et al., 2018b; 2019; Feng et al., 2019). With regard to the interaction of characteristics, the use of attention mechanisms in these methods can be treated as an explicit modelling of the interaction of characteristics while neglecting the modelling of implicit interactions of characteristics.

The methods mentioned above either model implicit and explicit feature interactions isolated or adopt a suboptimal way to combine them, which can be inefficient. In this work, we aim to address these problems by introducing a small MLP layer that dynamically generates kernels conditioned

by the current instance to capture both implicit and explicit feature interactions. The core idea is to first *generate* context weights and biases from the context stream, and then *aggregate* them with the input stream adaptively. We formulate a generic function and implement it with an efficient *dynamic parameterized operation* (DPO). The first weight generator projects contextual features into high-dimensional representation, which models implicit conditional bias. The second feature aggregator aims to fuse input features and projected contextual representation in a multiplicative way, e.g., matrix multiplication and convolution, maintaining both low- and high-order information.

For feature-based modeling, we introduce *feature-based DPO* where the weight-generate operation dynamically produces instance-wise filters conditioned on the embedded context. The feature-aggregate function then applies instance-wise filters to the flattened input by matrix multiplication, allowing to learn multiplicative features. In particular, we further propose a new class of DPO, called *field-based DPO*, which is not only instance-specific but also field-specific. In that case, the filters vary from field to field and from instance to instance, allowing more complex interactions along the field dimension.

For user behavior modeling, we introduce *sequence-based DPO* that consists of two variants: behavior-behavior dynamic operation and query-behavior dynamic operation. A representative method of dynamic convolution (Chen et al., 2020; Yang et al., 2019) shares the convolution kernel, which is generated by the global average of the inputs. Similarly, (Wu et al., 2019) proposed DyConv, a lightweight fine-grained convolution that depends only on time-step, reinforcing the encoder-based language modeling framework. However, our methods incorporate both local and global information as they jointly use locality-aware methods (e.g., convolution or separable convolution) followed by a global average pooling layer to produce instance-wise weights. The query-behavior dynamic operation is specialized designed for the decoder-based framework in CTR prediction, aiming to capture target-behavior dependency.

To our best knowledge, this is the first attempt to extend the business of dynamic neural networks to CTR prediction with extensive experiments on two fundamental scenarios. The comprehensive study against existing solutions validates the superiority of our proposed method. Moreover, we demonstrate that incorporating DPO into the real-world ranking system is beneficial.

Our contribution can be summarized as followed:

- We propose a generic formulation for capturing multiplicative interaction via weight-generate and feature-aggregate function, termed DPO.

- For feature-based modeling, we propose two variants, named field-based and feature-based DPO, offering a unifying view of implicit and explicit feature interaction. Decomposing these operations, we find they implicitly inherit low- and second-order representation.

- For user behavior modeling, we propose two sequence-based variants: behavior-behavior and query-behavior DPO. The first one computes locally perceptual dynamic filters and the second one learns target-behavior dependency in a multiplicative manner. We demonstrate that such operations can benefit the self-attention layers by higher computational efficiency through modeling locality as inductive bias.

- The proposed dynamic parameterized networks outperform state-of-the-art methods by a significant margin on both public and real-world production datasets. We also give a comprehensive study about the relationship of our proposed methods to previous Factorization Machine (Rendle, 2010) and CrossLayer (Wang et al., 2017). We further demonstrate the effectiveness and superiority of our method with an online A/B test in real-world applications by incorporating it into the fine-rank stage of the real-world ads system.

## 2 METHOD

We first review the mainstream approaches of feature-based and user behavior (sequence-based) modeling under the situation of CTR prediction[1]. After that, we introduce DPO and provide several specific instantiations designed for traditional feature-based and sequence-based modeling.

---

[1]Related work is in Appendix A due to space limitation.

### 2.1 PRELIMINARY

**Traditional CTR prediction** methods mainly predict a probability of a *user* click an *item*, which serves as a fundamental evaluation criterion for computing advertising systems. Typically, in a given scenario (the contexts), users click on certain items (item profiles) based on their own needs (query) and pautorferences (user profiles). Consequently, a model considers four fields of features, i.e., *query, user profile, item profile, and contexts* to predict:

$$CTR = F(query, user\ profile, item\ profile, contexts) \tag{1}$$

where item and user profiles contain up to tens of fine-grained static attributes depending on the specific circumstances.

**Sequence-based CTR prediction** involves user behaviors additionally:

$$CTR = F(query, user\ behavior, user\ profile, item\ profile, contexts) \tag{2}$$

where the models can learn from the behaviors that have occurred under certain contexts and query in the past to make judgments on the current items. As mentioned in KFAtt (Liu et al., 2020), the behavior module can be formulated as: $\hat{v}_q = UserBehavior(q, k_{1:T}, v_{1:T})$, where $k_{1:T}$ and $v_{1:T}$ are given $T$ historical clicked items and corresponding query words. The most used strategy is to adopt the the self-attention mechanism (Vaswani et al., 2017), which naturally learns multiplicative interaction between query and the historical behavior.

### 2.2 FORMULATION

Namely, multiplicative interaction (Jayakumar et al., 2020) has been proposed to fuse two different sources of information with the goal of approximating function $f_{target}(\boldsymbol{x}, \boldsymbol{z}) \in \mathbb{R}^c$, where $\boldsymbol{x}$ and $\boldsymbol{z}$ are the input and context respectively. Similarly, we give a generic formulation of DPO in CTR prediction task as:

$$y_i = \frac{1}{C(\boldsymbol{z})} \sum_{\forall j} f(\boldsymbol{x}_i; \boldsymbol{g}_i(\boldsymbol{z}_j; \boldsymbol{\theta})) \tag{3}$$

Here $i$ is the index of a position (in the field, or sequence), whose response is calculated with the generated output of $\boldsymbol{z}$ over all existing positions. $\boldsymbol{x}$ is the input embedding, while $\boldsymbol{z}$ denotes any specified context. The generate function $g$ aims to compute **dynamic weights and bias** followed as one of the inputs of the pairwise aggregate function $f$, which learns the interactive features reflecting the relationship between $\boldsymbol{x}_i$ and $\boldsymbol{z}_j$. The output is finally normalized by a factor $C(\boldsymbol{z})$.

MLP and convolution typically process input and context features in an additive way with fixed weights. While in Eqn. (3), using instance-wise generated weights and bias from contexts $z$, the additive nature is transformed to multiplicative. DPO is also different from bilinear layer (Lin et al., 2015; He & Chua, 2017) for Eqn. (3) computes representation based on the generated weights over all positions, whereas bilinear layer aggregates information over all positions between $\boldsymbol{x}$ and $\boldsymbol{z}$, leading to large memory consumption. Furthermore, our generated dynamic weights can maintain more local information, which complements the global counterpart, e.g., self-attention. DPO is a flexible block and can easily work together with MLP and self-attention layers.

### 2.3 FEATURE-BASED DPO

Given $\boldsymbol{x} \in \mathbb{R}^m$ and $\boldsymbol{z} \in \mathbb{R}^n$ as inputs and context, due to the lack of position information, the generic formulation degrades as $y = f(\boldsymbol{x}; g(\boldsymbol{z}; \boldsymbol{\theta}))$. For simplicity, we consider $f$ in the form of a linear transformation: $f(\boldsymbol{x}; \boldsymbol{z}) = W(\boldsymbol{z})\boldsymbol{x}$, where $W(\boldsymbol{z})$ is an instance-wise two-dimensional matrix generated by function $g$. Now, We discuss the choice of function $g$. Following the hypernetworks (Ha et al., 2016), a natural choice of $g$ is a fully-connected layer to form dynamic weights and bias:

$$y = \underbrace{(\hat{\boldsymbol{W}}^T \boldsymbol{z} + \hat{\boldsymbol{b}})^T}_{DyWeights} x + \underbrace{(\dot{\boldsymbol{W}}^T \boldsymbol{z} + \dot{\boldsymbol{b}})}_{DyBias} = \underbrace{\boldsymbol{z}^T \hat{\boldsymbol{W}} \boldsymbol{x}}_{explicit} + \underbrace{\dot{\boldsymbol{W}}^T \boldsymbol{z} + \hat{\boldsymbol{B}}^T \boldsymbol{x} + \dot{\boldsymbol{b}}}_{implicit} \tag{4}$$

where $(\hat{\boldsymbol{W}}, \hat{\boldsymbol{b}}, \dot{\boldsymbol{W}}, \dot{\boldsymbol{b}}) \in (\mathbb{R}^{n \times mc}, \mathbb{R}^{mc}, \mathbb{R}^{n \times c}, \mathbb{R}^c)$. However, the size of $\hat{\boldsymbol{W}}$ has quadratic space complexity, unsuitable for deployment in real-world application. Here, we consider a low-rank method in practice, e.g., a two-layer MLP:

$$g(z) = \boldsymbol{W}_2^T \sigma(\boldsymbol{W}_1^T z + \boldsymbol{b}_1) + \boldsymbol{b}_2 \tag{5}$$

Figure 1: Illustration of feature-based and field-based dynamic parameterized operation.

where $(\boldsymbol{W}_1, \boldsymbol{b}_1) \in (\mathbb{R}^{n \times l}, \mathbb{R}^l)$ and $(\boldsymbol{W}_2, \boldsymbol{b}_2) \in (\mathbb{R}^{l \times (mc+c)}, \mathbb{R}^{mc+c})$, $\sigma$ is a non-linear function. Then, we can decompose the output into explicit dynamic weights and bias. The right inductive bias depends on how we select context $z$ and $g$. We denote the complexity of $f$ is $O(mc)$ less than $O(mc + nc)$ of plain MLP layer, while $g$ scales up to $O(lmc + ln)$. To reduce the complexity, we set $l$ as a small number and use a multi-head mechanism (Vaswani et al., 2017).

**Relation to Cross Network**: A cross layer (Wang et al., 2017) take the feature interaction formulation as $\boldsymbol{x}_{i+1} = \boldsymbol{x}_0 \cdot \boldsymbol{x}_i \boldsymbol{w}_i + \boldsymbol{b}_{i+1} + \boldsymbol{x}_i$, where $\boldsymbol{x}_i \boldsymbol{w}_i$ is scalar. We prove CrossLayer is the simplest formulation of DPO. Let's take $\boldsymbol{x}_0$ as $z$, $\boldsymbol{x}_i$ as $x$ and only use 1-layer MLP as weight-generate function, whose hidden states are 1, (i.e. $\boldsymbol{z} = \boldsymbol{x}_0, \hat{\boldsymbol{W}} \in \mathbb{R}^n, \dot{\boldsymbol{W}} = 0, \hat{\boldsymbol{B}} = 1$ ). Thus, we get a scalar output of $g$ as the same as the multiplicative term of CrossLayer. In this way, DPO aims to imitate multiplicative operation.

## 2.4 FIELD-BASED DYNAMIC PARAMETERIZED OPERATION

Given $\boldsymbol{X} \in \mathbb{R}^{t_1 \times m}$ and $\boldsymbol{Z} \in \mathbb{R}^{t_2 \times n}$ as inputs and context, where $t_1$ and $t_2$ represent the field numbers respectively, our goal is modeling the interaction between $\boldsymbol{x}_i$ and $\boldsymbol{z}_j$ over all field positions. A simple idea is to treat field-based operation as multiple feature-based operations followed by summation over all output. Thus, Eqn. (3) can be expressed as $\boldsymbol{y}_i = f(\boldsymbol{x}_i; \frac{1}{C(\boldsymbol{z})} \sum_{\forall j} g_i(\boldsymbol{z}_j; \boldsymbol{\theta}))$, which means all fields share the same instance-wise weights.

However, the field-based operation interacts between all fields, which sometimes introduces unnecessary feature coupling (i.e., multiplicative interaction of brand ID and time, etc.). The empirical evidence finds over-coupling brings more noise and then results in underfitting, albeit their capacity of learning high-order features. A considerable method is to use Self-Field dynamic operation without heavily hand-crafted feature engineering, formulated as: $\boldsymbol{y}_i = f(\boldsymbol{x}_i; g_i(\boldsymbol{x}_i; \boldsymbol{\theta}))$ by removing cross-field interactions. Apart from Summation-based and Self-based methods, a more attentive solution can be used to aggregate the dynamic attributes: $\boldsymbol{y}_i = f(\boldsymbol{x}_i; \sum_{\forall j} h(\boldsymbol{z}_j; w_j) g_i(\boldsymbol{z}_j; \boldsymbol{\theta}))$, where $h$ is an attention layer. Beyond taking position into consideration, we can interact the whole context with inputs without explicit summation instead of concatenation, formulated as: $\boldsymbol{y}_i = f(\boldsymbol{x}_i; g_i([\boldsymbol{z}_1, \boldsymbol{z}_2..., \boldsymbol{z}_{f2}]; \boldsymbol{\theta}))$, where $[\cdot, \cdot]$ is a concatenation operation. These four methods learn pairwise field-based interaction from coarse to fine to model high-order representation, while the feature-based method combines both low- and high-order information over all fields. The complex weight-generate function can be designed for the right inductive bias, but we do not specifically consider such a method for online serving and leave it to future work.

**Relation to FM**: Here, we slightly modify the origin FM implementation (Rendle, 2010) as: $y = \sum_{\forall i} \sum_{\forall j > i} \boldsymbol{x}_i^T \boldsymbol{x}_j$ by removing the LR term, that takes interaction among all field positions into consideration. Given inputs and the context as $\boldsymbol{x}_i$ and $\{\boldsymbol{x}_j, \forall j \neq i\}$, the function $f$ is simply matrix multiplication and $g$ is the identity function, then Eqn. (3) can be decomposed to: $y_i = \frac{1}{t_1 - 1} \sum_{\forall j \neq i} \boldsymbol{x}_i^T \boldsymbol{x}_j$. Thus, FM can be viewed as the self-excluded version of field-based dynamic operation, where the context is other field features different to the input features.

## 2.5 SEQUENCE-BASED DYNAMIC PARAMETERIZED OPERATION

User behavior modeling focus on learning from their historical actions to predict whether the users click the current items. As a comparison, transformer-based solutions (Liu et al., 2020; Zhou et al., 2018a) explored the encoder-decoder framework to learn long-range dependencies both source-to-

Figure 2: Illustration of homogeneous behavior and heterogeneous query-behavior dynamic parameterized operation.

source and source-to-target, where the encoder exploits multi-head self-attention to extract session interest and the decoder aggregates the query-specific interest. Following the encoder-decoder framework, we consider two variants, i.e., homogeneous Behavior-Behavior and heterogeneous Query-Behavior dynamic operation (homo- and hetero- DPO). We show their multiplicative attributes on Appendix D.

**Homogeneous Behavior-Behavior DPO** aims to capture feature interaction at different time-steps of behavior. Given $X = Z \in \mathbb{R}^{t \times n}$ as inputs and context, where $t$ represents the behavior length. For user behavior modeling, our goal is to model the long- and short-term feature interaction. As mentioned above, a long-term function aims to learn non-local interaction between all positions while short-term ones only care about the local information. Thus, a natural way is to adopt global-aware weight-generate function $g$ and local-aware feature-aggregation $f$. Different to Section 2.3 and Section 2.4, we adopt convolution as $f$ which is widely used for modeling local sequence information with learned weights. For simplicity, we consider function $f$ in the form of a 1D-convolution with kernel size $k$, while feature-based and field-based methods only use MLP.

$$\boldsymbol{y}_i = f(\boldsymbol{x}_i; \frac{1}{C(\boldsymbol{z})} \sum_{\forall j} g(\boldsymbol{z}_j; \theta)) = \frac{1}{C(\boldsymbol{x})} \sum_{l=\lfloor -\frac{k}{2} \rfloor}^{\lfloor \frac{k}{2} \rfloor} \sum_{j=0}^{t} g_l(\boldsymbol{x}_j; \theta) \boldsymbol{x}_{i+l} \tag{6}$$

$$\sum_{j=0}^{t} g_l(\boldsymbol{x}_j; \theta) = \boldsymbol{W}_{2,l}^T \sigma(\boldsymbol{W}_{1,l}^T \sum_{j=0}^{t} \boldsymbol{x}_j + \boldsymbol{b}_{1,l}) + \boldsymbol{b}_{2,l} \tag{7}$$

Eqn. (6) shows the function $f$ can act as a convolution operation without bias term which models local neighborhood by dynamic weight, where $\boldsymbol{x}_{i-l}$ is the extracted behavior in position $i$. Eqn. (7) gives a instantiation of weight-generate function $g$. Firstly, we aggregate all sequence information and project them into a select operator $\boldsymbol{s} \in \mathbb{R}^d$ by $\boldsymbol{W}_{1,l}$ and $\boldsymbol{b}_{1,l}$, where $\boldsymbol{W}_{1,l} \in \mathbb{R}^{n*d}, \boldsymbol{b}_{1,l} \in \mathbb{R}^d$ and $\sigma$ is activation function. Secondly, we use $s$ to explicit aggregate expert weight, where $\boldsymbol{W}_{2,l} \in \mathbb{R}^{d \times (nc)}$ and $\boldsymbol{b}_{2,l} \in \mathbb{R}^{nc}$. To use dynamic depthwise-convolution, we can set $c = 1$. Eqn. (7) captures the multiplicative interaction correspond to global aggregation features. To further strengthen locality, we can adopt local-aware function to capture short-term information of context $\boldsymbol{x}$ (e.g. convolution, separable convolution etc.).

**Heterogeneous Query-Behavior DPO** aggregates all sequential behaviors as context targeting to interaction with query. Give $\boldsymbol{x} \in \mathbb{R}^m$ and $\boldsymbol{Z} \in \mathbb{R}^{t*n}$, we take function $f$ as linear transformation as mentioned in Section 2.3. Eqn. (3) learns interaction between query and behavior over all length followed by summation, and the simplest formulation can be derived as:

$$y = f(\boldsymbol{x}; \frac{1}{C(\boldsymbol{z})} \sum_{\forall j \in t} g(\boldsymbol{z}_j; \boldsymbol{\theta})) = g(\frac{1}{C(\boldsymbol{z})} \sum_{\forall j \in t} \boldsymbol{z}_j; \boldsymbol{\theta})^T \boldsymbol{x} \tag{8}$$

Similar to feature-based and field-based methods, query-behavior dynamic operation can easily learn rich multiplicative interaction and conditional inductive bias. The weight-generate function $g$ aims to learn the weight representation $\boldsymbol{W}_g \in \mathbb{R}^{m \times c}$. Typically, we can exploit a specific aggregation function, such as Eqn. (7). Compared to self-attention in decoder (Liu et al., 2020), DPO focuses attention on instance-weights based on context, while self-attention takes bipartite attention matrix to aggregate value units. Thus, we conjecture they are two orthogonal and complementary solutions.

Table 1: Comparison with different algorithms of feature-based datasets over 5-runs results. Std≈1e-3.

| Datasets | Movielens-tag | | Avazu | | Criteo | |
|---|---|---|---|---|---|---|
| Base Model | Auc | Logloss | Auc | Logloss | Auc | Logloss |
| FM (Rendle, 2010) | 0.9388 | 0.2797 | 0.7497 | 0.3740 | 0.7933 | 0.4574 |
| AFM (Xiao et al., 2017) | 0.9414 | 0.2804 | 0.7454 | 0.3766 | 0.7953 | 0.4554 |
| HOFM (Blondel et al., 2016) | 0.9410 | 0.3088 | 0.7516 | 0.3756 | 0.7960 | 0.4551 |
| NFM (He & Chua, 2017) | 0.9355 | 0.2955 | 0.7531 | 0.3761 | 0.7968 | 0.4537 |
| PNN (Qu et al., 2016) | 0.9469 | 0.2792 | 0.7526 | 0.3737 | 0.8026 | 0.4509 |
| CIN (Lian et al., 2018) | 0.9494 | 0.2600 | 0.7533 | 0.3756 | 0.8042 | 0.4472 |
| AFN (Cheng et al., 2020) | 0.9477 | 0.2753 | 0.7512 | **0.3731** | 0.8061 | 0.4458 |
| CrossNet (Wang et al., 2017) | 0.9323 | 0.2929 | 0.7498 | 0.3756 | 0.7915 | 0.4585 |
| CrossMix (Wang et al., 2020) | 0.9379 | 0.2934 | 0.7526 | 0.3738 | 0.8019 | 0.4490 |
| DNN | 0.9521 | 0.2576 | 0.7533 | 0.3745 | 0.8028 | 0.4483 |
| **Feature-based DPN** | **0.9535** | **0.2538** | **0.7556** | **0.3733** | **0.8097** | **0.4425** |
| **Field-based DPN** | 0.9507 | 0.2561 | **0.7536** | **0.3750** | **0.8049** | **0.4467** |

Table 2: Ablation study on MovieLens-tag dataset over 5-runs results. We show Auc and logloss.

(a) **Instantiations of weight-generate functions**: 1 feature-based dynamic operation of different $g$ is added into first layer of a 2-layer MLP(300-300). Std of metrics $\approx$ 1e-3.

| $g$ | Auc | Logloss | Params |
|---|---|---|---|
| Base (3-layer MLP) | 0.9471 | 0.2656 | 192k |
| Base (2-layer MLP) | 0.9521 | 0.2576 | 101k |
| Base (2-layer MLP, 400) | 0.9514 | 0.2568 | 175k |
| HyperDense | 0.9524 | 0.2715 | 371k |
| Eqn. (5), $l$=4, $\sigma$=sigmoid | 0.9522 | 0.2756 | 129k |
| Eqn. (5), $l$=4, $\sigma$=softmax | **0.9527** | **0.2563** | 129k |
| Eqn. (5), $h$=2, $\sigma$=softmax | 0.9515 | 0.2622 | 101k |
| $\boldsymbol{P}\phi(\boldsymbol{z})\boldsymbol{Q}$ | 0.9503 | 0.2609 | 108k |
| $\boldsymbol{W}_0 + \boldsymbol{P}\phi(\boldsymbol{z})\boldsymbol{Q}$ | 0.9520 | 0.2566 | 117k |

(b) **Layers and Context**: we compare the results by replacing fully-connected layer with 1 and 2 feature-based dynamic operation of the 2-layer DNN baseline. Also, we compare dynamic results of different context.

| Models, Eqn. (5) | Context | Auc | Logloss | Params |
|---|---|---|---|---|
| Base (2-layer MLP) | None | 0.9521 | 0.2576 | 101k |
| fc1 | $\boldsymbol{z}_0$ | 0.9527 | 0.2563 | 129k |
| fc2 | $\boldsymbol{z}_0$ | 0.9522 | 0.2582 | 372k |
| fc1 + fc2 | $(\boldsymbol{z}_0, \boldsymbol{z}_0)$ | 0.9532 | 0.2562 | 400k |
| fc1 + fc2 | $(\boldsymbol{x}_0, \boldsymbol{x}_0)$ | 0.9535 | **0.2538** | 400k |
| fc1 + fc2 | $(\boldsymbol{x}_0, \boldsymbol{y}_{l_0})$ | 0.9530 | 0.2549 | 401k |
| fc1 + fc2 | $(\boldsymbol{x}_t, \boldsymbol{x}_t)$ | 0.9533 | 0.2545 | 400k |
| fc1 + fc2 | $(\boldsymbol{z}_t, \boldsymbol{z}_t)$ | **0.9544** | 0.2566 | 400k |
| fc1 + fc2 | $(\boldsymbol{z}_m, \boldsymbol{z}_m)$ | 0.9530 | 0.2575 | 400k |

(c) **Instantiations of field-based dynamic parameterized network**: we compare different aggregation function for MoK, Eqn. (5) as shown in Table 2a in a two-layer field-based DPN. Also, we compare the parameters and time cost with feature-based method and implicit interaction models. We implement all models on 12 cores Intel(R) Xeon(R) CPU E5-2683-v4@2.10GHz with TensorFlow op.

| Model | context | Auc | Logloss | Params | CPU Second/Epoch |
|---|---|---|---|---|---|
| DNN | None | 0.9521±5e-4 | 0.2576±2e-3 | 101k | 44.7 |
| Field-DNN (implicit), 100-100 | None | 0.9445±1e-3 | 0.2669±4e-3 | 11k | 76.7 |
| Larger Field-DNN, 200-200 | None | 0.9473±8e-4 | 0.2663±4e-3 | 44k | 175.4 |
| Feature-based DPN | $(\boldsymbol{z}_t, \boldsymbol{z}_t)$ | 0.9544±6e-4 | 0.2566±4e-3 | 400k | 61.4 |
| Field-based DPN (Summation) | $(\boldsymbol{x}_0, \boldsymbol{y}_{l_0})$ | 0.9451±1e-3 | 0.2720±7e-3 | 46k | 44.2 |
| Field-based DPN (Self + implicit) | $(\boldsymbol{x}_0, \boldsymbol{y}_{l_0})$ | 0.9488±1e-3 | 0.2607±2e-3 | 57k | 86.4 |
| Field-based DPN (Summation + implicit) | $(\boldsymbol{x}_0, \boldsymbol{y}_{l_0})$ | 0.9495±1e-3 | 0.2595±1e-3 | 57k | 90.0 |
| Field-based DPN (Attention + implicit) | $(\boldsymbol{x}_0, \boldsymbol{y}_{l_0})$ | 0.9489±1e-3 | 0.2620±2e-3 | 57k | 93.7 |
| Field-based DPN (Concat + implicit) | $(\boldsymbol{x}_0, \boldsymbol{y}_{l_0})$ | 0.9507±1e-3 | 0.2561±2e-3 | 87k | 95.5 |

## 3 EXPERIMENTS

We perform comprehensive experiments on feature modeling and user behavior modeling of public and real-world production CTR prediction datasets.

### 3.1 EXPERIMENTS ON FEATURE MODELING

**Setting.** We evaluate with MovieLens-tag, Criteo, Avazu with following questions:

- How does DPN perform (effectiveness and efficiency) compared with other base models?

- How do different contexts and weight-generate functions influence the performance?

We use AUC and Logloss as metrics for public datasets. For all experiments, we evaluate the effectiveness of baseline models with the same training setting in AFN (Cheng et al., 2020) implemented by TensorFlow (Abadi et al., 2016). We adopt Adam (Kingma & Ba, 2014) as optimizer with best searched learning rate of a batch size 4096 for all models. We fix the embedding ranks as 10 across all datasets and use same deep neural network (e.g., 3 layers MLP of 400-400-400) with Batch Normalization and ReLU (Ioffe & Szegedy, 2015; Nair & Hinton, 2010) if without specifically noted. The details of our proposed methods are described in Appendix B.

**Comparison with state-of-the-art results.** Table 1 shows the results from AFN (Cheng et al., 2020) and our reimplemented results in the same setting. We note all these methods are single model without DNN components. First, our feature-based DPN consistently achieves better performance than other explicit interaction methods and also the implicit DNN baseline, which confirms the dynamic aspects contributes to implicit feature interaction. Additionally, when the train dataset and features get larger, the overwhelming margin get larger (e.g. 0.13% on MovieLens-tag, 0.23% on Avazu, 0.69% on Criteo), showing promising potential ability for applied in real-world production. Secondly, our field-based DPN perform better than the other explicit interaction module. We note field-based methods models the relation over different attributes (i.e., UserID, MovieID etc.) where low- and high-order information are captured in a totally different way. Specially, field-based DPN obtain additive module in parallel with multiplicative one while other high-order interaction methods follow an opposite stacked framework to learn the multiplicative features (Qu et al., 2016; Cheng et al., 2020; He & Chua, 2017).

**Effectiveness of different instantiations of weight-generate function.** Table 2a compares different types of a feature-based dynamic operation added to the DNN baseline (right after the embedding layer for replacing the fully-connected layer). After we search the best DNN baseline model, we replace a dynamic operation with the first fully-connected layer. We list the results of different weight-generate functions, where not all methods perform better than the baseline. We implement the hypernetworks-based idea (Ha et al., 2016) as HyperDense which slightly improve the baseline while add a big chunk of computation resulting for optimization difficulty. When we adopt our proposed simple and effective method as shown in Eqn. (5), gate mechanism can be exploited for better performance, which means mixture of kernels have better generality. Furthermore, we explore some approaches to reduce the complexity of $g$, such as Multi-Head mechanism, Matrix decomposition. However, they instead downgrade the performance even cannot compete on par with the baseline. We provide more ablation study on the Appendix E.

**Multi-Layer Feature-based DPN with different contexts.** Table 2b shows the results of deeper dynamic parameterized network with different context. We separately replace DPO with first, second, and all fully connected layers in 2-layer MLP. Table 2b shows that more feature-based DPO in general lead to better results regardless of context. We argue multiple feature-based operations can learn rich and high-order dynamic interaction by imitating MLP. High-order message can be processed with non-linear function layer by layer, which is hard to be found useful via multiplicative models.

In Table 2b, we also study the effectiveness of different context. We set the flatted inputs $x_0$ as the inputs of DPN and evaluate the performance of different contexts such as $x_0$ and $z_0$ where $x_0$ and $z_0$ is the flattened outputs of inputs and context embeddings respectively. We found they share similar results for most experiments while getting the best performance when we set the context as $z_t$ (i.e. use the tag information of context embeddings as context inputs). Under careful selection of hyperparameters, this best result mainly originates from the expert knowledge of MovieLens dataset and recommendation system. Meanwhile, it reveals a nice property of our methods: the intrinsic decoupling attribute can be more separably modeled. Interestingly, we find our methods improve results of the infrequent user on MovieLens datasets, as shown in Table 3. We believe the dynamic interaction can warm up the infrequent user embeddings, demonstrating the potential of our methods for the cold-start problem.

**Effectiveness and Efficiency of Field-based DPNs.** We design a family of field-based dynamic networks aiming to capture atomic relationships among fields' features. Table 2c presents results of different aggregate function. We find that all dynamic operations with different context aggregate func-

Table 3: Results of infrequent user movielens datasets where occur times of a user is less than 20.

| Model | context | Auc | Logloss |
|---|---|---|---|
| DNN (implicit) | None | 0.9386±1e-4 | 0.2956±1e-2 |
| Feature-based DPN | $(z_t, z_t)$ | **0.9411**±8e-4 | 0.2911±4e-3 |
| Feature-based DPN | $(z_m, z_m)$ | 0.9399±5e-4 | 0.2929±2e-3 |
| Feature-based DPN | $(z_u, z_u)$ | 0.9408±5e-4 | 0.3000±5e-3 |

Table 4: Adapt sequence-based DPO (sDPO) on Transformer. We evaluate the effectiveness of combination of multi-head self-attention mechanism and sDPO.

| Encoder | | Decoder | | |
|---|---|---|---|---|
| MSA | Homo | MSA | Heter | AUC |
| ✓ | ✗ | ✓ | ✗ | 0.8718 |
| ✓ | ✓ | ✓ | ✓ | 0.8755 |
| ✓ | ✓ | ✓ | ✗ | 0.8728 |
| ✓ | ✗ | ✓ | ✓ | 0.8809 |
| ✓ | ✗ | ✗ | ✓ | 0.8731 |
| ✗ | ✓ | ✓ | ✗ | 0.8775 |
| ✗ | ✓ | ✗ | ✓ | **0.8849** |

Table 5: Comparison with state-of-the-art on Amazon dataset for user behavior modeling. We record the mean AUC over 5 runs. We mainly compare our methods with a well-known attention mechanism.

| Model | All | New | inFreq |
|---|---|---|---|
| DIN (Zhou et al., 2018b) | 0.8292 | 0.8029 | 0.7937 |
| DIEN (Zhou et al., 2019) | 0.8675 | 0.8457 | 0.8375 |
| Trans (Vaswani et al., 2017) | 0.8718 | 0.8522 | 0.8438 |
| KFAtt (Liu et al., 2020) | 0.8789 | 0.8578 | 0.8496 |
| DIN + Heter | **0.8836** | 0.8554 | **0.8583** |
| DIEN + Heter | 0.8693 | 0.8476 | 0.8414 |
| Trans + Heter | 0.8809 | 0.8608 | 0.8526 |
| Trans + Homo | 0.8728 | 0.8530 | 0.8440 |
| sDPN | **0.8849** | **0.8615** | **0.8590** |

tions perform better than the static component, even only correspond to themselves.

We may hypothesize that additional contexts can benefit the field features after having been processed for imitating linear transformation, containing multiplicative interaction between inputs and contexts. However, we find the time-consuming is worrying in the CPU machine when the dimensions of outputs are relatively large, making it venturesome to be applied on real-world production.

## 3.2 Experiments on User Behavior modeling

**Experiment setting.** We evaluate sequence-based DPN (sDPN) on Amazon Electronics Datasets. We only adopt AUC as metrics with the same training setting in KFAtt (Liu et al., 2020) implemented by TensorFlow. The task is to predict whether a user will write a review for the target item after reviewing historical items. We refer the readers to KFAtt (Liu et al., 2020) for more details.

**Comparison with state-of-the-art.** From Table 5 we can see that sequence-based DPN achieves the best performance than all state-of-the-arts on total situations, including the strong baseline KFAtt. When incorporating heterogeneous DPO into the attention mechanism, we find both DIN (Zhou et al., 2018b) and DIEN (Zhou et al., 2019) perform better than the origin baseline where the armed DIN outperform all the base models on Amazon datasets by a large margin, which shows that heterogeneous DPO can effectively learn complementary representation which can benefit the attention mechanism.

**Adaptation to Self-Attention.** Table 4 presents us how sDPO performs when incorporated with self attention mechanism. For homogeneous DPO, we find it performs slightly better than MHSA counterparts. When we use session-wise representation for user behavior modeling, self-attention can capture local information over handcraft scope of time designed by experts while narrow neighbor interaction of convolution may not contribute to learning users' attention due to the short session. For heterogeneous DPO, we find it can effectively facilitate the decoder counterpart no matter which encoder we adopt. Overall, the sequence-based DPN(sDPN) achieve best results than other combination, which shows the effectiveness of our propose homogeneous and heterogeneous DPO.

## 3.3 Experiments on Real-world Production dataset and Online A/B Testing

We conduct all feature-based, field-based, and sequence-based experiments on the Real-world Production dataset. In the offline experiments, we observe the progressive improvement from modern rank models consisting of advanced user behavior and multi-modal features model in our advertising system. Table 6 shows our **DyMLP** and **DyJoint** get significant advancement compared to origin implementation. For feature interaction modeling, **DyMLP** shows better performance than the commonly used DNN component while only add a little extra cost. Despite we don't explore the effectiveness of ensemble DPN models in a public dataset, Table 6 presents even one lightweight field-based DPO that can benefit the generality. To reduce the complexity, we use a multi-head mechanism in feature-based DPN which may influence the effectiveness.

Table 6: Results on Real-world Production dataset. We show the details of how to incorporate DPNs into online ads system in Appendix G.1. For feature modeling, we name DPNs as DyMLP. For both feature and user behavior modeling together with online model, we name DPNs as DyJoint.

| Module | MLP | Feature | Field | KFAtt | Homo | Heter | Auc(+gain) | Throughput (batch/s) |
|---|---|---|---|---|---|---|---|---|
| Base | ✓ | ✗ | ✗ | ✗ | ✗ | ✗ | 0.7523 | 101 |
| DyMLP | ✗ | ✓ | ✗ | ✗ | ✗ | ✗ | 0.7530(↑0.07) | 101 |
|  | ✗ | ✓ | ✓ | ✗ | ✗ | ✗ | **0.7550**(↑0.27) | 88 |
| Online | ✓ | ✗ | ✗ | ✓ | ✗ | ✗ | 0.7598 | 55 |
| DyJoint | ✗ | ✓ | ✓ | ✓ | ✗ | ✗ | 0.7609(↑0.11) | 50 |
|  | ✗ | ✓ | ✓ | ✓ | ✓ | ✗ | 0.7618(↑0.20) | 48 |
|  | ✗ | ✓ | ✓ | ✓ | ✗ | ✓ | 0.7624(↑0.26) | 48 |
|  | ✗ | ✓ | ✓ | ✓ | ✓ | ✓ | **0.7633** (↑0.35) | 46 |

For sequence-based DPO, we conduct more ablation studies in Appendix G.3. Our homogeneous DPO can act as a specific form like dynamic convolution. Incorporating it with a session-based self-attention encoder, we can inject inductive bias learned from local neighborhood information into global long-term dependencies based on Transformer-like models (Vaswani et al., 2017; Feng et al., 2019). Beyond that, heterogeneous DPO can learn more conditional multiplicative interaction which models the user interest on given items, showing greater power than the homogeneous component. Combined with all techniques, we get the best results by a large margin to the online model.

Table 7: Results of Online A/B testing.

| Model | CTRgain | eCPMgain | TP99 latency |
|---|---|---|---|
| Online | 0% | 0% | 24ms |
| DyJoint | ↑1.0% | ↑1.2% | 29ms |

In the online A/B test, Table 7 shows DyJoint contributes 1.0% CTR gain against the online component, which demonstrates the superiority over the highly optimized base model on our ad system. However, DyJoint leads to larger online latency compared to the base model due to the increment of model parameters and memory access.

## 4 CONCLUSION

In this paper, we describe a new class of neural networks that captures both explicit and implicit interaction via dynamic parameterized operation. Our proposed block can be easily inserted into existing CTR predicting architectures for fusing features from different modalities. Our experiments show that it overwhelms the existing feature-crossing-based and attention-based models on two fundamental tasks. Furthermore, we confirm its representation effectiveness in the real-world production dataset. For the theoretical understanding, we decouple dynamic operation for comprehensive study with high-order feature-crossing methods and self-attention. Overall, we open a new era where current mainstream solutions are dominated by self-attention mechanisms and MLP in CTR prediction.

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

Table 8: Statistics of datasets for feature modeling.

| Datasets | instance | fields | features |
|---|---|---|---|
| Criteo | 45302405 | 39 | 2086936 |
| Avazu | 40428967 | 22 | 1544250 |
| Movielens-tag | 2006859 | 3 | 90445 |
| Movielens-1M | 739012 | 7 | 3529 |
| Real-world Production | 12 Billion | 96 | N/A |

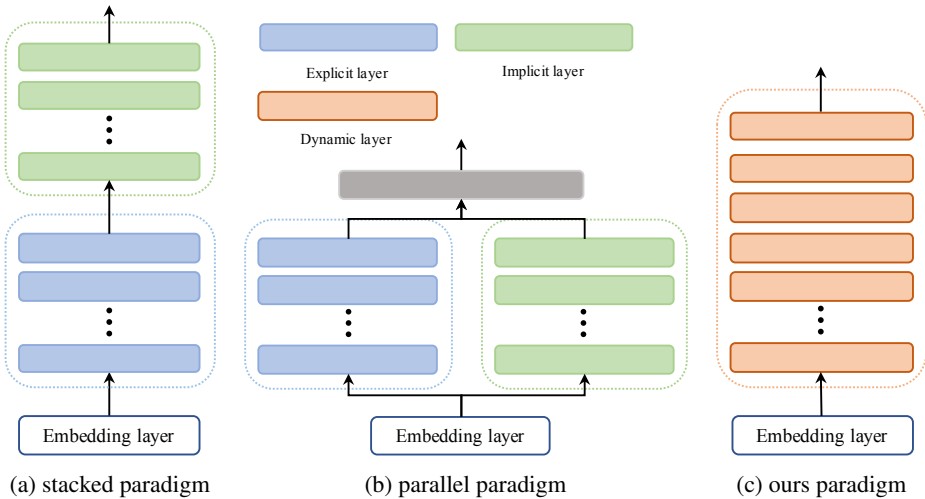

Figure 3: Illustration of stacked, parallel and ours paradigm for traditional CTR prediction.

# A   RELATED WORK

**Feature Crossing as high-order interaction** has been widely explored in CTR prediction and recommendation system by capture a cross feature which can be defined as $\prod_{i=0}^{n} x_i^{a_i}$. Factorization Machine (Rendle, 2010) was firstly proposed to calculate similarity by using **inner product** between two different features (e.g, item feature and user feature) borrowed from the collaborative-filtering based model (Sarwar et al., 2001). With the rise of deep neural network, recent works focus on combining feature crossing with *Embedding+MLP* paradigm for better performance (i.e. stacked and parallel paradigm) as shown in Fig. (3a) and Fig. (3b). Late fusion is mostly used in **parallel paradigm**. Wide and deep (Cheng et al., 2016) propose wide component to form order-2 features by cross-product transformation between sparse inputs and a deep layer for order-1 features. DCN (Wang et al., 2017) propose to use parallel structure composed by bit-wise **cross-layer** and deep neural network. DeepFM (Guo et al., 2017) adopts a parallel structure to fuse the FM and DNN outputs. To facilitating the high-order representation, xDeepFM (Lian et al., 2018) are proposed to generate vector-wise high-order feature, achieving further advancement by stacking multiple interaction layers. Contrast to polynomial networks, AutoInt(Song et al., 2019) exploit a multi-head self-attention layer to automatically learn the high-order feature interaction of all atom features(e.g. item id, user id, brand id etc.). Ensembled with DNN, AutoInt achieve better result. Contrast to parallel ensemble methods, **stacked paradigm** uses an explicit layer to extract cross features followed by an implicit layer to process them, i.e. DNN. NeuFM (He & Chua, 2017) uses Bi-Interaction layer with stacked deep neural network for bridging generalized linear models and deep learning. AFN (Cheng et al., 2020) models cross features of adaptive orders followed by 3-layer MLP. Despite the inherit strong generality incorporated with DNN, stack paradigm often resorts to further ensemble with DNN for better results.

However, due to either the low capacity or high latency of so-called interaction layer, those methods typically failed in real-world rank system. Recent application of polynomial networks on CTR prediction can be seen as a special case of multiplicative interaction to formulate scalar-wise, element-

wise or feature-wise multiplier fusion between projected embeddings. We prove that our methods can degrade as the bit-wise cross-layer (Wang et al., 2017) and pairwise interaction layer (Rendle, 2010).

**User behavior modeling** aims to extract users' dynamic and evolving interest over items. Earlier works (Covington et al., 2016; Song et al., 2016; Yu et al., 2016) use a target-independent manner to aggregate total behaviors, failed to learn the interaction between target and past behaviors. Recent works (Zhou et al., 2019; 2018b;a; Liu et al., 2020) achieve massive improvement by adopting attention mechanism for modeling long-range representation between user historical behaviors and other target features. Orthogonally, our proposed methods can be formulated as multiplicative interaction between two different streams.

**Dynamic Neural Networks** enjoy preferable properties than static ones, whose computation graph and forward parameters are fixed with limited representation power (Han et al., 2021). Instance-wise dynamic neural networks typically have dynamic structure (Tanno et al., 2019), dynamic weights (Yang et al., 2019; Ha et al., 2016; Chen et al., 2020), or dynamic layers (Huang et al., 2018) in inference time. Jayakumar (Jayakumar et al., 2020) provide theoretical comprehension about why *dynamic weights* are beneficial: *HyperNetworks* (Ha et al., 2016) can be thought of generalised multiplicative interaction in the non-factorised sense, whose filters are generated by conditional inputs.

# B  IMPLEMENTATION DETAILS OF DPN

## B.1  DATASETS AND EXPERIMENTAL PROTOCOLS

**Public dataset for feature-interaction modeling**. We conduct experiments on three public real-world datasets in Section 3.1 as shown in Table 8. **MovieLens-tags**[2] has been prepared into (user, movie, tag) format for personalized recommendation. Totally, there are 2006859 tuples, constituting of 3 categorical fields: UserID, MovieID and tag. **Avazu**[3] is click-through rate dataset including 22 feature fields of user features and advertisement attributes. There are about 40 million instances in total. **Criteo**[4] is a benchmark dataset for displayed ads CTR prediction, containing 26 categorical fields and 13 numeric fields. It has about 45 million user-click records on displayed ads. We split the dataset in 7:2:1 for training, validataion and testing respectively.

**Public dataset for user behavior modeling** We conduct sequence modeling on Amazon book dataset (McAuley et al., 2015) to learn users' interest. We use the 5-core Electronics subset, including 1689188 instances with 192403 users and 63001 goods from 801 categories. The task is to predict the intent of writing users' review for a target item regard to the historical reviews. In practice, we use the last review of each user as test split while others as train split. The negative instances are randomly sampled from not-reviewed items.

**Real-world Production dataset** is a daily traffic log generated from the search advertising system. Typically, we sample negatively a first 32-days click-through logs for training data containing 2.4 billion samples and non-negatively the next-day click-through logs for test data containing about 1 million samples. For the user-behavior data setting, we follow the KFAtt (Liu et al., 2020) to choose user clicks/queries in previous 70 days as behavior $v/k$. The other features consist of the query, user and ad profile, ad image and contexts, built for modeling feature interaction.

**Metrics**: AUC is almost the default offline evaluation metrics in advertising system due to the direct relation to online performance. Thus, we use *AUC* as a evaluation metric for evaluation on both public dataset and real-world production dataset. For fair comparison, we additionally add *Logloss* as another metric when evaluated on public dataset. We denote that 0.001-level improvement of *AUC* on CTR prediction task is significant, which has been a common sense in previous works (Cheng et al., 2016; Guo et al., 2017; Lian et al., 2018; Wang et al., 2020; Wang et al., 2017).

---

[2]https://grouplens.org/datasets/movielens/

[3]https://www.kaggle.com/c/avazu-ctr-prediction

[4]https://www.kaggle.com/c/criteo-display-ad-challenge

## B.2 Embedding Layer

For all ctr models, the categorical features and dense features (UserID, ItemID, User Age etc) are hashed to sparse one-hot features as the inputs of embedding layers. We project i-th one-hot feature from sparse high-dimensional space to a lower one via $\boldsymbol{x}_{emb,i} = \boldsymbol{W}_i \boldsymbol{e}_i$, where $e_i \in \{0,1\}^v$ and $\boldsymbol{W} \in \mathbb{R}^{e \times v}$; $v$ and $e$ means vocab and embedding size respectively. Thus, for **feature-based** modeling, we take the output of the concatenation of all the embedding vectors: $x_0 = [x_{emb,0}, x_{emb,1}, ..., x_{emb,n}]$ as our named **features**. For **field-based** modeling, we instead take the embedding vectors of every sparse ID as fields aiming to learning the interaction of pairwise features (e.g. [UserID, ItemID], [UserID, UserAge]).

## B.3 Feature-Based and Field-Based Modeling

As mention in Eqn. (3), all of our methods induce an extra information features $z$ for boosting generalization. However, we don't use extra fields as show in Table 8 which represents the effectiveness comes from the combination of fully-connected layer and feature-crossing module. For feature-based and field-based DPN, we stack several feature- and field-based DPO layer followed by classifier layer to build them. The implicit interaction module of feature-based DPN is fully-connected layer with non-linear function, while the implicit part of field-based DPN has two parts: field-interaction module $\boldsymbol{W}_f$ and linear transformation module $\boldsymbol{W}_l$, where $\boldsymbol{W}_f \in \mathbb{R}^{n \times n}$ and $\boldsymbol{W}_l \in \mathbb{R}^{e_{l-1} \times e_l}$. Thus the static outputs can be calculate by $\boldsymbol{W}_f \boldsymbol{X}_0 \boldsymbol{W}_l$, where $\boldsymbol{X}_0 \in \mathbb{R}^{n \times e_{l-1}}$. We name it as **Field-DNN**.

## B.4 User Behavior Modeling on Amazon Dataset

To evaluate the effectiveness of adapting sDPO on various methods, we adpot different strategy to combine sDPO with DIN, DIEN and Transformer. For DIN and DIEN, we produce the heterogeneous DPO outputs followed by element-wise summation with origin attention mechanism outputs. For Transformer, we sum the homogeneous and heterogeneous DPO outputs and self attention outputs in encoder and decoder respectively. Except for the increment benefit over other methods, we test the effectiveness only using DPO, i.e. sequence-based Dynamic Parameterized Network (sDPN). The sDPN simply replaces the self attention counterparts on Transformer framerwork by DPO layers.

## B.5 User Behavior Modeling for Real-world Production Dataset

We follow the setting of KFAtt (Liu et al., 2020) along with the feature-based methods and field-based methods, evaluated only on real-world production dataset. There, 96 multi-modal features are first embedded into 16-dimension vectors and then fed for field-based and feature dynamic operation. The origin implement make use of 4-layer MLP with dimension 1024, 512, 256, 1. When there is a 30 minutes' time interval between adjacent behaviors, we conduct a session segmentation. For each instance, we use at most 10 sessions and 25 behaviors per session. The learnt hidden user interest, $\hat{v}^q \in R^{64}$ is concatenated to the output of 1st FC layer together with a 150-dimensional visual feature vector. Our homogeneous behavior-behavior operation are used to fuse the session-based self-attention encoder output in additive way with kernel size = 3. Our heterogeneous query-behavior operation are used to fuse the predicted interest produced by cross-attention. All the weight generate function are a SE-layer (Hu et al., 2018) whose intermediate down ratio is grid searched from [0.125, 0.25, 0.5].

## C   DETAILS OF THE RELATION TO FM

As shown in (Rendle, 2010), FM can be formulated as:

$$
\begin{aligned}
y(x) &= b + \sum_{i=1}^{n} w_i x_i + \sum_{i=1}^{n} \sum_{j=i+1}^{n} <\boldsymbol{v}_i, \boldsymbol{v}_j> x_i x_j \\
&= linear(x; \boldsymbol{\theta}) + \frac{1}{n-1} \sum_{i=1}^{n} (\boldsymbol{v}_i x_i \otimes \sum_{\forall j != i} \boldsymbol{v}_j x_j) \\
&= linear(x; \boldsymbol{\theta}) + \frac{1}{n-1} \sum_{i=1}^{n} f(\boldsymbol{x}_i; g(\sum_{\forall j != i} \boldsymbol{x}_j))
\end{aligned}
\tag{9}
$$

Where $\boldsymbol{v}$ means the embedding table, $x$ mean the categorical features. Thus we can decompose the output $y$ as summation of each field-wise interaction features. Thus the function $f$ and function $g$ can be represented for matrix multiplication and identity function. The self-excluded version means we use the $\sum_{\forall j != i} \boldsymbol{x}_j$ as context features which exclude the input feature $\boldsymbol{x}_i$.

## D   ANALYSIS MULTIPLICATIVE ATTRIBUTE OF SDPO

### D.1   ANALYSIS OF HOMOGENEOUS DPO

$$
\begin{aligned}
\boldsymbol{y}_i &= f(\boldsymbol{x}_i; \frac{1}{C(\boldsymbol{z})} \sum_{\forall j} g(\boldsymbol{z}_j; \theta)) \\
&= \frac{1}{C(\boldsymbol{x})} \sum_{l=\lfloor -\frac{k}{2} \rfloor}^{\lfloor \frac{k}{2} \rfloor} \sum_{j=0}^{t} g_{l,w}(\boldsymbol{x}_j; \theta) \boldsymbol{x}_{i+l} + \sum_{j=0}^{t} g_{l,b}(\boldsymbol{x}_j; \theta) \\
&= \sum_{l=\lfloor -\frac{k}{2} \rfloor}^{\lfloor \frac{k}{2} \rfloor} (\boldsymbol{W}_{2,l}^T (\boldsymbol{W}_{1,l}^T \sum_{j=0}^{t} \boldsymbol{x}_j + b_{1,l}) + \boldsymbol{b}_{2,l})^T \boldsymbol{x}_{i+l} + g_b(\hat{\boldsymbol{x}}) \\
&= \sum_{l=\lfloor -\frac{k}{2} \rfloor}^{\lfloor \frac{k}{2} \rfloor} (\sum_{j=0}^{t} \boldsymbol{x}_j^T \boldsymbol{W}_{1,l} \boldsymbol{W}_{2,l} \boldsymbol{x}_{i+l} + \boldsymbol{W}_{s,l}^T \boldsymbol{x}_{i+l}) + g_b(\hat{\boldsymbol{x}})
\end{aligned}
\tag{10}
$$

As shown in Eqn. (10), we show the homogeneous DPO actually learns both multiplicative and additive relation among local neighbor and each behavior features followed by summation aggregate function when we don't use non-linear function. To further reduce complexity, a common way is to learn the feature interaction between global embedding aggregated firstly and local neighbor, formulated as:

$$
\boldsymbol{y}_i = \sum_{l=\lfloor -\frac{k}{2} \rfloor}^{\lfloor \frac{k}{2} \rfloor} (\boldsymbol{W}_{2,l}^T \sigma(\boldsymbol{W}_{1,l}^T \sum_{j=0}^{t} \boldsymbol{x}_j + b_{1,l}) + \boldsymbol{b}_{2,l})^T \boldsymbol{x}_{i+l}
\tag{11}
$$

where $\sigma$ can be sigmoid and softmax function.

## D.2 ANALYSIS OF HETEROGENEOUS DPO

$$
\begin{aligned}
y &= f(\boldsymbol{x}; \frac{1}{C(\boldsymbol{z})} \sum_{\forall j \in t} g(\boldsymbol{z}_j; \boldsymbol{\theta})) \\
&= g_w(\frac{1}{C(\boldsymbol{z})} \sum_{\forall j \in t} \boldsymbol{z}_j; \boldsymbol{\theta})^T \boldsymbol{x} + g_b(\frac{1}{C(\boldsymbol{z})} \sum_{\forall j \in t} \boldsymbol{z}_j; \boldsymbol{\theta}) \\
&= (\boldsymbol{W}_2^T (\boldsymbol{W}_1^T \sum_{j=0}^{t} \boldsymbol{z}_j + \boldsymbol{b}_1) + \boldsymbol{b}_2)^T \boldsymbol{x} + g_b(\hat{\boldsymbol{z}}) \\
&= (\boldsymbol{W}_2^T (\boldsymbol{W}_1^T \hat{\boldsymbol{z}}))^T \boldsymbol{x} + (\boldsymbol{W}_2^T \boldsymbol{b}_1 + \boldsymbol{b}_2)^T \boldsymbol{x} + g_b(\hat{\boldsymbol{z}}) \\
&= \underbrace{\hat{\boldsymbol{z}}^T \boldsymbol{W}_1 \boldsymbol{W}_2 \boldsymbol{x}}_{query-whole} + \boldsymbol{W}_s^T \boldsymbol{x} = \frac{1}{C(\boldsymbol{z})} \sum_{j=0}^{t} (\underbrace{\boldsymbol{z}_j^T \boldsymbol{W}_1 \boldsymbol{W}_2 \boldsymbol{x}}_{query-each}) + \boldsymbol{W}_s^T \boldsymbol{x} + g_b(\hat{\boldsymbol{z}})
\end{aligned}
\tag{12}
$$

As shown in Eqn. (12), We decompose the simplest formulation of Eqn. (8) where $\boldsymbol{W}_1 \in \mathbb{R}^{d*n}$, $\boldsymbol{W}_2 \in \mathbb{R}^{n*(m*c)}$, $\boldsymbol{b}_1 \in \mathbb{R}^n$, $\boldsymbol{b}_2 \in \mathbb{R}^{mc}$, $\boldsymbol{x} \in \mathbb{R}^m$, $\boldsymbol{z} \in \mathbb{R}^n$. The equation omits the reshape operation. We can see that Heterogeneous DPO captures multiplicative and additive between each behavior and query different to the self attention mechanism. However, Eqn. (12) can't be enhanced by non-linear function, Empirically, a better way is to learn the relation of the embedding representing for the whole behavior sequence and query features, which further reduce the computations and also have strong compatibility with complex weight-generation function $g$ such as SE-layer (Hu et al., 2018).

## E ADDITIONAL EXPERIMENTS ON MOVIELENS-TAG

### E.1 ABLATION STUDY

We perform ablations on variants of feature-based and field-based DPN. Unless specified otherwise, all experimental results in this sections report MovieLens AUC and logloss by training a DPN architecture that replace weighted fully-connected layer.

**Varing depth and width for feature-based and field-based DPN** Table 9 presents the impact of different depth and width on performance. Our experiments indicate that Feature-based DPN can achieve better performance with similar time consuming over a wide range hyperparameters, which demonstrate the robustness of the method. However, the atomic field-based DPN still lack the capacity to compete on par with MLP baseline. Furthermore, we find the atomic fields' interaction is hard to inert into feature-based DPO due to the extra fields dimension.

**MD, MH vs Plain** Table 10 presents the performance of matrix decomposition and multi-head mechanism on feature-based DPN. Despite their success when applied in other domains (Vaswani et al., 2017; Li et al., 2021; Wu et al., 2019), we find these methods don't work in traditonal CTR prediction. We conjecture the multi-head mechanism hurts the expressivity of fully-connected layer due to the separable interaction among channels. For matrix decomposition, we aim to generate instance-wise kernels by $\boldsymbol{P}\phi(\boldsymbol{z})\boldsymbol{Q}$, where $\boldsymbol{P} \in \mathbb{R}^{n \times r}$, $\phi(\boldsymbol{z}) \in \mathbb{R}^{r \times r}$, $\boldsymbol{Q} \in \mathbb{R}^{r \times m}$. We set $r = 32$ in Table 10. Thus, we may solve the low-rank completion problem for better performance. We conjecture simply utilization of dynamic parameterized fully-connected layer by matrix decomposition leads to optimization difficulty. We leave it into future study.

## F ADDITIONAL EXPERIMENTS ON MOVIELENS-1M

Besides experiments on three public datasets, We extraly conduct feature-based and field-based ablation studies on widely-used MovieLens-1M.

**MovieLens-1M** is a relative small dataset (Harper & Konstan, 2015), contains users' rating on movies with 7 features, e.g. user features and movie features and rating. We follow the experiment

Table 9: Comparison of different depth and width over 5-runs results on MovieLens-tag.

| Model | Auc | Logloss | Auc(infreq) | Logloss(infreq) | Params | CPU Time/epoch |
|---|---|---|---|---|---|---|
| DNN, 100-100 | 0.9498±6e-4 | 0.2607±3e-3 | 0.9358±5e-4 | 0.2986±3e-4 | 13k | 35s |
| DNN, 100-100-100 | 0.9468±4e-4 | 0.2648±1e-3 | 0.9324±9e-4 | 0.2983±2e-4 | 24k | 37s |
| Fe DPN, MoK, 100-100 | **0.9516**±3e-4 | 0.2581±4e-3 | **0.9389**±8e-4 | 0.2951±1e-2 | 53k | 39s |
| DNN, 200-200 | 0.9507±9e-4 | 0.2575±2e-3 | 0.9370±1e-3 | 0.2945±4e-3 | 47k | 38s |
| DNN, 200-200-200 | 0.9476±5e-4 | 0.2633±2e-3 | 0.9336±1e-3 | 0.2962±4e-3 | 88k | 41s |
| Fe DPN, MoK, 200-200 | **0.9533**±9e-4 | **0.2527**±8e-4 | **0.9412**±9e-4 | **0.2897**±2e-3 | 187k | 48s |
| DNN, 64-64 | 0.9487±2e-4 | 0.2619±1e-3 | 0.9341±1e-4 | 0.2984±3e-3 | 6k | 35s |
| Fe DPN, MoK, 64-64 | 0.9502±1e-3 | 0.2593±3e-3 | 0.9370±2e-3 | 0.2959±5e-3 | 25k | 37s |
| Fi DPN, Sum, MoK, 32-32 | 0.9414±7e-4 | 0.2796±1e-3 | 0.9268±1e-3 | 0.3164±3e-3 | 6k | 40s |
| Fi DPN, Sum, MoK, 64-64 | 0.9428±1e-3 | 0.2777±3e-3 | 0.9280±1e-3 | 0.3119±6e-3 | 20k | 40s |
| Fi DPN, Sum, MoK, 128-128 | 0.9453±1e-3 | 0.2723±2e-3 | 0.9309±1e-3 | 0.3103±3e-3 | 73k | 51s |

Table 10: Variants of Feature-based DPN over 5-runs results on MovieLens-tag.

| Model | Head | Auc | Logloss | Auc(infreq) | Logloss(infreq) | Params | CPU Time/epoch |
|---|---|---|---|---|---|---|---|
| DNN, 200-200 | | 0.9507±9e-4 | 0.2575±2e-3 | 0.9370±1e-3 | 0.2945±4e-3 | 47k | 38s |
| Fe DPN, MoK, 100-100 | 1-1 | **0.9516**±3e-4 | **0.2581**±4e-3 | **0.9389**±8e-4 | 0.2951±1e-2 | 53k | 39s |
| Fe DPN, MoK, 100-100 | 2-1 | 0.9498±4e-4 | 0.2618±2e-3 | 0.9379±1e-3 | **0.2913**±2e-3 | 44k | 43s |
| Fe DPN, MoK, 100-100 | 1-2 | 0.9513±5e-4 | 0.2583±2e-3 | 0.9382±5e-4 | 0.2963±5e-3 | **23k** | 40s |
| Fe DPN, MoK, 100-200 | 1-2 | 0.9511±7e-4 | 0.2591±2e-3 | 0.9385±6e-4 | 0.2958±4e-3 | 34k | 42s |
| Fe DPN, MoK, 100-100 | 2-2 | 0.9479±9e-4 | 0.2657±2e-3 | 0.9353±1e-3 | 0.2974±4e-3 | 14k | 39s |
| Fe DPN, MoK, 200-200 | 2-2 | 0.9491±1e-3 | 0.2719±1e-2 | 0.9380±1e-3 | 0.2994±1e-2 | 48k | 43s |
| Fe DPN, MD, 100-100 | 1-1 | 0.9467±1e-3 | 0.2708±4e-3 | 0.9330±1e-3 | 0.3007±5e-3 | 20k | 41s |
| Fe DPN, MD, 200-200 | 1-1 | 0.9494±5e-4 | 0.2628±2e-3 | 0.9362±6e-4 | 0.2975±4e-4 | 30k | 46s |

setting of AutoInt (Song et al., 2019), which randomly split 80% of dataset into train data, 10% into valid data and the rest 10% into test data. And we set the ratings below 1s and 2s as 0s, above 4s and 5s as 1s. All rating 3s should be removed.

**Training** For ablation study on MovieLens-1M, our baseline models follow the settings in AutoInt (Song et al., 2019) with random initialized weights, implemented by Tensorflow (Abadi et al., 2016) with the tricks (Srivastava et al., 2014; Nair & Hinton, 2010) but we don't use BN (Ioffe & Szegedy, 2015) due to the poor performance. The feature-based model take the same setting as DeepCrossing (Shan et al., 2016) introduced in AutoInt with 4-layer MLP of size 100. The function $g$ is 2-layer MLP. The field-based models use three dynamic layer and set hidden states as 64 the same as their public code[5], (e.g. replace self-attentive layer with our field-based dynamic operation). After that, we use grid search to select hyperparameters, (i.e. dropout rate, learning rate). For fair comparison, all methods set embedding dimension as 16 and use Adam (Kingma & Ba, 2014) as default optimizer with $\beta_1 = 0.9$, $\beta_2 = 0.999$ , batch size is 1024. However, we only replace feature-based operation on the final two layers before classifier, due to the memory limitation and insert 1-layer field-based operation firstly.

**Comparison with state-of-the-art** Table 12 shows the results from AutoInt (Song et al., 2019). Due to the random data splits, we rerun their experiments for comparison. We note the 0.001-level fluctuation on evaluation metric compared to the cited' ones is acceptable when trained and evaluated on randomly split setting. Nevertheless, our feature-based method surpasses the other existing high-order based methods (Wang et al., 2017; Lian et al., 2018; He & Chua, 2017) by a good margin, even the proposed field-based operation by ourselves. However, what surprise us is the heavily tuned DNN model still excels high-order parts in contrast to the results reported in the AutoInt. We infer that MLP-based DNN model can be a good universal approximate function to attain a suitable local minimum with good hyperparameters. In this way, there is no doubt feature-based solution is better than others, because of its capacity for learning both implicit and explicit feature interaction. As far as we know, our proposed method is state-of-the-art solution in the same setting, even without the DNN component.

---

[5]https://github.com/DeepGraphLearning/RecommenderSystems/tree/master/featureRec

Table 11: Hyper-parameters of Dynamic Parameterized Network in MovieLens-1M.

| Hyper-parameter | feature-based | field-based |
|---|---|---|
| fields | MovieID, zipcode, Age, Occupation, Gender, Timestamp | |
| embedding size | 16 | 16 |
| hidden size | 64 | 64 |
| $g$ | 2-layer MLP | 2-layer MLP |
| $\sigma$ | sigmoid | sigmoid |
| residual | False | True |
| LN | False | False |
| non-linear | Relu | Relu |
| initializer | glorot | glorot |
| context | $x_{l-1}$ | $x_{l-1}$ |
| Layers | 2 | 2 |
| Heads | 1 | 1 |
| Learning rate | 3e-4 | 1e-3 |
| Adam $\epsilon$ | 1e-8 | 1e-8 |
| Adam $\beta_1$ | 0.9 | 0.9 |
| Adam $\beta_2$ | 0.999 | 0.999 |
| Batch size | 1024 | 1024 |
| Params | 86k | 30k |

Table 12: Comparison with different algorithms over 10-runs results. Std≈1e-3. ↓ means below while ↑ means above. Throughput means training time of one epoch. We also cite the results from AutoInt (Song et al., 2019).

| Model | Auc | Logloss | Params |
|---|---|---|---|
| MLP(ours) | 0.8475 | 0.3785 | 53k |
| DeepCrossing | 0.8448 | 0.3814 | N/A |
| AutoInt(ours) | 0.8448 | 0.3812 | 39k |
| NFM | 0.8357 | 0.3883 | N/A |
| CrossNet | 0.7968 | 0.4266 | N/A |
| CIN | 0.8286 | 0.4108 | N/A |
| Feature-based | **0.8522** | **0.3726** | 86k |
| Field-based | 0.8452 | 0.3806 | 30k |

**Comparison with different heads in feature-based operation**   Table 13a compares different head numbers of 4 dynamic parameterized layers with context $z$ as genre. We find a good head number can effectively improve the performance, while larger lead to bad results. We derive this large gap is mainly due to the reason our baseline model use relative smaller embedding size and hidden state with increasing head numbers. When head numbers equal to 4, the size of total parameters is about 11k, which seriously downgrades the performance. However, a properly head number trades off the extent of implicit and explicit high-order interaction, inducing larger variance of selection on dynamic weights/bias.

**Comparison with different non-linear function in feature-based operation**   Table 13b shows the performance of different non-linear function in Eqn. (5). Here, we set $l = 4$ and head numbers equal to 2, for simplicity. We find even using plant low-rank approximation without non-linear function, the result still better than high-order part reported in Table 1. This gives evidence that non-restrict multiplicative interaction among hidden states can be a complementary component for further improving the capacity of MLP. When using non-linear function *sigmoid* and *softmax*, we found the performance moves a significant step about 0.001-level. Constrained non-linear function controls the outputs of selection layer between $(0, 1)$, which can be seen as the attention over kernels (e.g. mixture of expert layers). We note that the best non-linear function depends on the dataset respectively. Such methods for kernels aggregation are friendly for optimization, allowing us to learn diversity of weights while consume nearly same time as plain fully-connected layer.

Table 13: Ablation study of feature-based dynamic operation

(a) Comparison with different head number. $l$=4, $\sigma$ is sigmoid, context is genre.

| head | Auc | Logloss |
|------|------|---------|
| 1 | 0.8483 | 0.3781 |
| 2 | 0.8504 | 0.3738 |
| 4 | 0.8136 | 0.4126 |

(b) Comparison with non-linear function. $l$=4, $h$=2, context is genre.

| $\sigma$ | Auc | Logloss |
|----------|------|---------|
| identity | 0.8494 | 0.3781 |
| sigmoid | 0.8504 | 0.3738 |
| softmax | 0.8506 | 0.3745 |

(c) Comparison with different context. $l$=8, $h$=2, $\sigma$ is softmax.

| context | Auc | Logloss |
|---------|------|---------|
| genre | 0.8511 | 0.3741 |
| gender | 0.8457 | 0.3782 |
| genre, gender | 0.8452 | 0.3800 |
| all | **0.8522** | **0.3726** |

Table 14: Ablation study of field-based dynamic operation

(a) Comparison with different aggregation methods.

| Agg | Auc | Logloss |
|-----|------|---------|
| Summation | 0.8372 | 0.3868 |
| Self | 0.7917 | 0.4264 |
| Attention | **0.8452** | **0.3806** |
| Concat | 0.8443 | 0.3808 |

(b) Comparison with different combination methods of field-based and feature-based operation.

| Methods | structure | Auc | Logloss |
|---------|-----------|------|---------|
| Self + Feature | stacked | 0.8471 | 0.3787 |
| Self (One layer) + Feature | stacked | 0.8520 | 0.3738 |
| Self + Feature | parallel | **0.8532** | 0.3725 |
| Weighted Sum + Feature | stacked | 0.8527 | 0.3746 |
| Weighted Sum + Feature | parallel | **0.8536** | 0.3729 |

**Comparison with different context in feature-based operation** Table 13c compares the different context of feature-based dynamic operation. We can see that different combination of fields as context $z$ will lead to variant results, where the worst only improve slightly than AutoInt, but less than DNN model. Interestingly, we select separately the user feature $gender$ and movie feature $genre$ as context. The gender typically consist two classes of male and female. Thus, the multiplicative interaction can be understood as how the gender influences whether a user rates a movie. Contrast to human intuition on gender, the results reveal that gender is not main reason for different rating on movie, e.g. explicit modeling gender interaction with other features leads to unnecessary bias. However, when modeling with genre of movies, we get large improvement on performance compared to the genders'. We conjecture the genre plays important role on rating of users. Moreover, when combined with all field features, we can achieve best results, representing the given features can be classified effectively in high dimension.

**Comparison with different aggregation function in field-based operation** The feature-based methods give empirical evidence that how to select head numbers, non-linear function and context. For simplicity, we choose the same best setting in ablation study of field-base dynamic operation. Table 14a compares four methods of how to aggregate differnet field information as mentioned in Section 2.4. When only using the field to correspond itself, we find it perform worst due to the scarcity of field-wise feature interaction compared to the simple summation version. When using a weighted-sum layer and concatenation-MLP layer, we find such methods improve almost 0.005-level compared to summation-based solution. The main difference is using a learned linear combination of all field features, which implicit models low-order feature interaction. Thus, we can conclude implicit linear transformation overall fields is beneficial for modeling high-order representation. However, we notice the DNN result overwhelm the field-wise operations', meaning the multiplicative interaction brought by high-order component is not the main course in CTR prediction.

**Does field-wise component contribute to feature-wise component?** Despite the low capacity of field-wise dynamic operation, we explore the contribution when combining field-wise component with feature-wise's. Table 14b shows there are no obvious benefit when we use both operation at same time no matter how we stack them, while facilitating the performance when composed of parallel structure. The stack structure directly exploit the field-based features as the inputs of feature-based layers. Once the fundamental representation has low diversity and expressivity, subsequent fancy modules can't perform their proper ability, e.g. three layers stacked Self-based operation dominates the effectiveness. However, we find the aggregation methods of weighted summation can effectively boost the performance sightly no matter which structure we used. Thus, we confirm

Table 15: Experiment Results on Real-world Production dataset

| DyMLP | SelfAtt | KFAtt | Encoder (k = 3) | Decoder | Auc |
|:---:|:---:|:---:|---|---|---|
| ✓ | ✓ | ✓ | Sep + SE + Homo | Sep + SE + Heter | 0.7633 |
| ✓ | ✓ | ✓ | Sep + SE + Homo, k = 5 | Sep + SE + Heter | 0.7626 |
| ✓ | ✓ | ✓ | SE + Homo | SE + Heter | 0.7622 |
| ✓ | ✓ | ✓ | Conv1D | Sep + SE + Heter | 0.7624 |
| ✓ | ✓ | ✓ | Conv1D, k = 5 | Sep + SE + Heter | 0.7624 |
| ✓ | ✗ | ✓ | Sep + SE + Homo | Sep + SE + Heter | 0.7617 |
| ✗ | ✓ | ✓ | Self + MLPs + Homo, k=1 | Heter | 0.7604 |

when combined those two methods in a suitable way, field-wise component can contribute to the feature-wise component for further advancement of performance.

# G ADDITIONAL EXPERIMENTS ON REAL-WORLD PRODUCTION DATASET FOR USER BEHAVIOR MODELING

## G.1 INCORPORATING DYNAMIC OPERATION INTO ONLINE RANKING SYSTEM

In previous section, we mainly discuss the dynamic mechanism among different inputs and context. However, real-world industry system scales behavior-based and feature-based modeling containing many techniques to precisely extract user interest and maintain low latency at the same time. We now introduce the whole behavior modeling module and feature modeling module deployed in our CTR prediction system, which consists of three parts: a MLP-based deep neural network, a transformer-based (Vaswani et al., 2017) encoder aims to learn long-range dependency of session-based behaviors, and a KFAtt-based (Liu et al., 2020) decoder to predict user interest along with total user actions. We term the encoder-decoder part as **DyTrans**, the MLP part as **DyMLP** and the whole module as **DyJoint**, followed by:

- Replace MLP with feature-based dynamic operation
- Insert field-based dynamic operation before MLP
- Combine Behavior-Behavior dynamic operation with Self-Attention
- Combine Query-Behavior dynamic operation with KFAtt

## G.2 IMPLEMENT DETAILS

**Encoder: Within Session Interest Extractor** KFAtt (Liu et al., 2020) mainly adopt the multi-head self-attention used in Transformer (Vaswani et al., 2017). While this self-attention is only conduct in session-based behavior for efficiency. Typically, we divide the behavior sequence into Sessions according to their occurring time. The self-attention can be formulated as:

$$MultiHead(K_s, K_s, V_s) = Concat(head_1, \ldots, head_h)W_O \qquad (13)$$

$$head_i = Attention(K_s W_i^Q, K_s W_i^K, V_s W_i^V) = softmax(K_s W_i^Q W_i^{K^T} K_s^T / \sqrt{d_k}) V_s W_i^V \qquad (14)$$

Incorporated our proposed methods, the module can be formulated as:

$$MultiHead(K_s, K_s, V_s) + DynamicOp(K_{s,k}, g_s(K_s)) \qquad (15)$$

**Decoder: Query-specific Interest Aggregator** KFAtt acts as the decoder to aggregate interest from all sessions for query-specific interest prediction. Incorporated with query-behavior decoder, it can be formulated as:

$$v^q = Concat(head_1, \ldots, head_h)W_O + DynamicOp(query, g(K)) \qquad (16)$$

### G.3    ABLATION STUDY

**Effect of varying dynamic kernels in encoder**    In session-based behavior modeling, we investigate the kernel size of homogeneous behavior operation which influences the temporal receptive fields. The larger kernel size means the larger locality information. Table 15 shows when we use larger dynamic kernel size, there exists marginally performance downgrading. The main reason is the session-based behaviors are not longer than 10, i.e. the locality has been learned by self-attention (Vaswani et al., 2017).

**Effect of convolution generation function**    Table 15 investigates the benefits of convolution-based weight-generate function. The separated convolution with k=3 shows further advancement in user behavior modeling. In this experiment, the weight-generate function and aggregation function both captures neighborhood representation, in order to learn short-term behaviors to short-term behaviors relationships, besides the dimension-behavior association.

**Effect of dynamic compared to static**    Table 15 also shows dynamic convolution with plain static convolution. As we can see, the dynamic operation generalize better than static one. But the locality brought by convolution still works in self-attention mechanism. When we decrease the kernel size as 1 without locality information and global context, we find it hardly improve the performance. We conjecture that such method can act as plain MoE over time-steps, demonstrating interaction among neighbor behaviors facilitates global connected features while features based on behaviors themselves only highlight the diagonal of attention matrix.

## H    UNDERSTANDING OF DYNAMIC PARAMETERIZED OPERATION

Our dynamic parameterized operation is one method to fuse two different stream by pretending ordinary *"matrix multiplication"* or *"Convolution"* with contextual kernels. In atom scene, we outspread the context representation from $\mathbb{R}^n$ to $\mathbb{R}^{m*c}$. As mentioned in section 2.3, when $c = 1$ in the simplest situation, the dot production can be used to calculate similarity(Huang et al., 2020; Rendle et al., 2020). Thus, we caluulate $c$ times similarity if we set *sigmoid* as non-linear function to learn robust features on the hypothesis that single calculation is not precise, i.e. $\sigma(z^t x) \to \sigma(z^t W x)$, also can be seen as bilinear operation(Lin et al., 2015) . Hence, the next dynamic layer aims to integrate refined similarity into another context, yielding high-order similarity. Without the dynamic bias term, DPO can be reduced to deep bilinear model. Beyond its capacity for channel interaction, DPO shows instance-wise interaction between channels and time-steps, i.e. convolve extracted time-steps with produced kernels by channels. Although fancy weight-generate function is not discussed in our work, we have shown it can introduce strong inductive bias, e.g. attention mechanism and local interaction.

**Relation to Dynamic Convolution**: We denote behavior-behavior operation coincides with the ideas of Dynamic Convolution(Han et al., 2021; Chen et al., 2020; Yang et al., 2019; Li et al., 2021) while differ on the motivation. DPO can be degraded to DyConv(Wu et al., 2019), if we set $g$ to combine softmax-normalized experts only depend on time-steps for depth-wise convolving.

**Relation to Mixture of Experts**: We denote it can act as another homogeneous implementation of MoE(Shazeer et al., 2017; Jacobs et al., 1991) where the weight-generate function can be armed with attention mechanism. Despite the similar complexity, our formulation is more favorable to explain the multiplicative interaction between two different features with an inner aggregation approach. However, Moe resorts to weighted aggregate the outputs of highly abstract expert towers.

**Relation to Multiplicative Interaction**: Eqn.(4) shows dynamic parameterized operation can be decompose as explicit feature interaction term and implicit feature interaction term. When $\hat{W}$ is a 3D Tensor, the explicit term can be simply seen as bilinear fusion operation (Lin et al., 2015), which captures channel interaction. The implicit term means a low-order feature interaction mechanism.

