# OpenReview forum: "Dynamic Parameterized Network for CTR Prediction"
_ICLR.cc/2022/Conference — ICLR 2022 Submitted_

### Official Review · Reviewer_2XAa · 2021-10-31

**Correctness:** 4
**Technical Novelty And Significance:** 3
**Empirical Novelty And Significance:** 2
**Recommendation:** 6
**Confidence:** 4

**Main Review:**

Strengths:
1. A good way to model and utilize feature/context interactions in order to better predict CTR.

2. Promising offline/online results and detailed discussion of offline results.

3. Well written paper. Clear illustration of the methodology as well as discussions about the relations with existing methods

Weaknesses:
1. Table 7 is way too simple, and so are related descriptions for the experiments. Are there more metrics to demonstrate the superiority of the new method? Is it affecting different segments of queries/users evenly? More details would be helpful to make the A/B test results stronger. Also, it seems CPC also increased. Any intuitions?

2. Why TP99 jumped a lot? Is it purely from the network structure change? How about TP50/90/99.9?

3. Page 4 "Relation to FM" would be more interesting to have details. The current high-level discussion is confusing. What is "self-excluded version"?

4. In section 2.1 Preliminary, why is it helpful to separate traditional/sequence-base CTR prediction?

5. To distinguish from "a random paper which tweaks the neural network structure for better performance", I would suggest more intuitions or discussions on why the added interaction component helps with the CTR prediction accuracy.


**Summary Of The Paper:**

The paper proposed a new method to handle feature/context interactions within the eCTR prediction neural networks. Details are provided in different scenarios. Experiments are conducted for offline and real-world experiments with promising results.

**Summary Of The Review:**

It's a methodology paper working on an important problem (CTR prediction). The proposed method is not extremely innovative but the idea is natural and makes sense, so are the results positive and promising. The novelty is a subjective matter but personally I see enough contributions to clarify and materialize some of the vague commonsense in the industry. One minor concern is about its relevance to "ICLR" which focuses on representation learning.

---

> ### Author Response · Authors · 2021-11-11
> **Response to Reviewer 2XAa**
>
> We thank the reviewer for highlighting the significance of our work and valuable suggestions. We address the reviewer’s questions below:
> ### Response to Question about the metrics in Table 7 in DPN:
>
> 1. For sponsored search advertising, we mainly focus on the CTR gain on the main traffic. More metrics like CVR, CPC, ctrPR, ctrPI, RPM etc have been attached below.
> | CVRgain | CPC    | ctrPR  | ctrPI  | RPM    |
> | ------- | ------ | ------ | ------ | ------ |
> | +1.22%  | +0.51% | +1.21% | +0.54% | +1.73% |
> 2. In the online A/B testing, typically it’s hard to evenly segment the traffic by user/query but the system can segment similar user/query for fair comparison. For the details, we adopt the 10% of the main traffic for online A/B testing where 5% serves on the base model and another 5% serves on ours. The system controls to segment the traffic evenly.
> 3. We observe that CPC has increased somewhat. Because the search advertisement system is more complicated than the recommendation system. After finerank stage, other modules will utilize the ranking result or the CTR values. For instance,  CVR models are mainly based on the CTR value to adaptively change the bid. However, the details of other modules are not revealed in finerank stage. Over-descriptive may affect the subject of our paper.
>
> ### Response to Question about TP99 latency:
> 1. The main reason is the computation brought by sequence-based dpo. Dynamic Convolution aims to build shared convolution kernels over all time-steps. However, TensorFlow doesn't provide an efficient implementation, which has occurred in DyConv[1]. Besides,  we find that the convolution operation induces large computation and memory overhead while the matmul in SelfAttention is light-weight. For most users, the modelled behavior length is more than 250 in the ad system which is longer than other online A/B test settings as far as we know. Thus we observe a similar gain on TP50/TP99/TP99.9 latency. A considerable method to address the computation complexity is to use multi-head sequence-based dpo or reduce the dimension in dpo while the memory overhead need more engineering works.
>
> ### Response to the Question about the details of "Relation to FM":
>
> As shown in [2], FM can be formulated as:
>
> \begin{equation}\begin{aligned}
>         y(x) &= b + \sum_{i=1}^{n} w_i x_i + \sum_{i=1}^{n} \sum_{j=i+1}^{n} <v_i, v_j> x_i x_j \\
>              &= linear(x;\theta) + \frac{1}{n-1} \sum_{i=1}^{n} (v_i x_i \otimes \sum_{\forall j!=i} v_j x_j)\\
>             %  &= linear(x;\theta) + \frac{1}{n-1} \sum_{i=1}^{n} g(x_i|\theta_g) h(\hat x_j|\theta_h)) \\
>              &= linear(x;\theta) + \frac{1}{n-1} \sum_{i=1}^{n} f(v_i x_i;g(\sum_{\forall j!=i } v_j x_j)) \\
>     \end{aligned}\end{equation}
> Where $v$ denotes the embedding table, $x$ denotes the categorical features. Thus we can decompose the output $y$ as summation of each field-wise interaction features. Thus the function $f$ and function $g$ can be represented for matrix multiplication and identity function. The self-excluded version means we use the $\sum_{\forall j!=i}v_j x_j$ as context features which exclude the input feature $v_i x_i$.
>
> ### Response to Question about the reason to separate traditional/sequnce-based CTR prediciton:
>
> Traditional CTR prediction mostly focuses on the feature-interaction methods, which neglect the sequential information of user behavior not only in datasets but also the proposed methods, e.g. DCN, DeepFM, FFM. Furthermore, when using user behavior, the sparse sequence cannot be incorporated into some specific models, like OPNN, AFN, DCN etc, due to the different behavior length after flattening and concatenation. Thus the sparse user behaviors typically are mapped into low dimensional embedding vectors and transformed into fixed-length features. However, sequence-based CTR prediction aims to capture user interests by well-designed sequential models to learn the sequence information of user behaviors, e.g. DIN, DIEN. In real-world ranking systems, sequence-based models are often built as a sub-module together with traditional CTR models.
>
>
> [1] Pay Less Attention with Lightweight and Dynamic Convolutions, ICLR 2019
>
> [2] Factorization Machines, ICDM 2010

---

> > ### Author Response · Authors · 2021-11-11
> > **Response to Reviewer 2XAa (con'd)**
> >
> > ### More intuitions and discussions:
> > We would like to clarify the relation to mixture of experts, multiplicative interaction and dynamic convolution for better understanding. we have concluded these discussions in our revised version.
> >
> > >Relation to Mixture of Experts: We denote that it can act as another homogeneous implementation of MoE where the weight-generate function can be armed with an attention mechanism. Despite the similar complexity, our formulation is more favorable to be explained as the multiplicative interaction between two different features with an inner aggregation approach. However, Moe resorts to weighted aggregate the outputs of highly abstract expert towers.
> >
> > >Relation to Multiplicative Interaction: Eqn.(4) in the paper shows dynamic parameterized operation can be decomposed as explicit feature interaction term and implicit feature interaction term. The explicit term can be simply seen as a bilinear fusion operation which captures channel interaction. The implicit term means a low-order feature interaction mechanism.
> >
> > >Relation to Dynamic Convolution: We denote behavior-behavior operation that coincides with the ideas of Dynamic Convolution while differing on the motivation. DPO can be degraded to DyConv, if we set $g$ to combine softmax-normalized experts only depending on time-steps for depth-wise convolving.
> >
> > >Understanding of dynamic parameterized operation: Our dynamic parameterized operation is one method to fuse two different stream by pretending ordinary "matrix multiplication" or "Convolution" with contextual kernels. In the atom scene, we outspread the context representation from $\mathbb R^{n}$ to $\mathbb R^{m*c}$. As mentioned in section 2.3, when $c=1$ in the simplest situation, the dot production can be used to calculate similarity. Thus, we calculate $c$ times similarity if we set sigmoid as nonlinear function to learn robust features on the hypothesis that single calculation is not precise, i.e. $\sigma(z^tx)\rightarrow \sigma(z^tWx)$, also can be seen as a bilinear operation . Hence, the next dynamic layer aims to integrate refined similarity into another context, yielding high-order similarity. Without the dynamic bias term, DPO can be reduced to a deep bilinear model. Beyond its capacity for channel interaction, DPO shows instance-wise interaction between channels and time-steps, i.e. convolve extracted time-steps with produced kernels by channels.
> > Although fancy weight-generate function is not discussed in our work, we have shown it can introduce strong inductive bias, e.g. attention mechanism and local interaction.

---

### Official Review · Reviewer_pP6H · 2021-11-01

**Correctness:** 3
**Technical Novelty And Significance:** 2
**Empirical Novelty And Significance:** 2
**Recommendation:** 5
**Confidence:** 4

**Main Review:**

Pros:
1) This paper is organised well and clearly written. 2) The idea of behavior modeling is novel to me. 3) Detailed theoretical analysis.

Cons:
1) Even though I think the proposed method is technically sound, the results cannot convince me. For the experimental results shown in Table 1, some baselines run worse than their normal performances, and the result of Dynamic Parameterized Network seems not good enough, e.g. compared with the results in [1].
2) Eq.5, how to demostrate the rationality of using the low-rank strategy.
3) few typos, e.g., section 2.5, actions.to

[1] Field-wise Learning for Multi-field Categorical Data, NeurIPS 2020.

**Summary Of The Paper:**

The paper proposed a module DPO for CTR prediction to enhence the explicit and implicit information. The authors claimed that they provide the first attempt to extend the dynamic neural networks to CTR prediction, and experiments show that DPN (Dynamic Parameterized Network) significantly outperforms other state-of-the-art methods.

**Summary Of The Review:**

The idea is somewhat novel, but the result is not good enough.

---

> ### Author Response · Authors · 2021-11-11
> **Response to Reviewer pP6H**
>
> Thank the reviewer for highlighting the novelty of our work and suggestions. We address the reviewer’s questions below:
>
> ### Question 1:
> Why are the results of Table 1 different to the results in [1]?
> ### Response to Q1:
> The main reasons are the different training settings which we would clarify below.
> 1. As shown in DPN, the experiment setting follows the same training setting in previous works( AFN[2], NFM[3]). They provide a unified train/valid/test splits on Avazu and Criteo. However, [1] uses random splits which is different to our setting.
> 2. Training Details:
>   * [1] reimplemented baseline models in PyTorch and select the best strategies on validation sets, especially the embedding dimension,  weight lambda for regularization term. For example they chose the embedding dimension from {20, 40, 60, 80, 100} for FM while {2,4,8,16} for FFM which is unfair to compare the model component. Furthermore, they randomly split the dataset into train (80%), valid(10%), test(10%). However, we split the dataset into train (70%), valid(20%), test(10%). Please refer to reading Appendix B in our paper.
>    * For fair comparison, we set the same embedding dimension as 10 for all models and without using l2 regularization. We only tune the dropout ratio and learning rates. Also different from [1], we implement all models in TensorFlow the same as AFN[2] and NFM[3].
>
>
> ### Question 2:
> Why do we use low-rank strategies?
> ### Respose to Q2:
> The original hypernetwork-style implementation causes larger computation and parameters. As shown in Table 2(a) in our paper, HyperDense achieves similar performance with MLP while having much more parameters.  In section 2.3, we have stated the low-rank methods help reduce the quadratic complexity of W. In Dynamic Conv[1] and CondConv[2], the low-rank strategy can be seen as the efficient implementation of MoE.
>
> The related part in original paper is quoted as follows:
>
> >  compares different types of a feature-based dynamic operation added to the DNN baseline (right after the embedding layer for replacing the fully-connected layer). After we search the best DNN baseline model, we replace a dynamic operation with the first fully-connected layer. We list the results of different weight-generate functions, where not all methods perform better than the baseline. We implement the hypernetworks-based idea as HyperDense which slightly improve the baseline while add a big chunk of computation resulting for optimization difficulty. When we adopt our proposed simple and effective method, gate mechanism can be exploited for better performance, which means mixture of kernels have better generality.
>
>
>
> Regarding the typo, thank you for pointing it out. We have modified it in the revised version.
>
>
>
> [1] Field-wise Learning for Multi-field Categorical Data, NeurIPS 2020.
>
> [2] Adaptive Factorization Network: Learning Adaptive-Order Feature Interactions
>
> [3] Neural Factorization Machines for Sparse Predictive Analytics
>
> [4] Dynamic Convolution: Attention over Convolution Kernels
>
> [5] CondConv: Conditionally Parameterized Convolutions for Efficient Inference

---

> > ### Comment · Reviewer_pP6H · 2021-11-11
> > **Concerns on experiment setting**
> >
> > In my experience, the split settings train (80%), valid(10%), test(10%) or train (70%), valid(20%), test(10%) will not make results changing a lot in the datasets Avazu and Criteo. I also checked some baselines used in this paper, such as [1]  and [2], both of them used the random split setting 80%, 10%, 10%, For example: in [2]:For the Criteo dataset and the Dianping dataset, we randomly split instances by 8:1:1 for training , validation and test; and in [1]: For each dataset, we randomly split the instances by 8:1:1.  However, in this paper, the author seems  to use the same setting (hyperparameters) for their baselines but using the different dataset splits instead, which is unfair. Furthermore, I suggest authors do not use a unified train/valid/test splits because it against the generality. I also suggest adding comparisons with [3].
> >
> > [1]Adaptive Factorization Network: Learning Adaptive-Order Feature Interactions
> > [2]xDeepFM: Combining Explicit and Implicit Feature Interactions for Recommender Systems
> > [3] Field-wise Learning for Multi-field Categorical Data, NeurIPS 2020.

---

> > > ### Author Response · Authors · 2021-11-17
> > > **Response to Experimental Setting**
> > >
> > > We re-run our DPN following the same setting as[1]. After simply tuning some parameters, we find our DPN perform competively compared to the results in [1].
> > >
> > > | Dataset | Models | Setting                                                | Auc    | Loss   | Params |
> > > | ------- | ------ | ------------------------------------------------------ | ------ | ------ | ------ |
> > > | Criteo  | DPN  | ebd_dim=100, lr=0.1, wdcy=1e-6, dropout=0.2            | 0.8130 | 0.4390 |  40.69M |
> > > | Avazu   | DPN  | ebd_dim=100, lr=0.01, wdcy=1e-5, dropout=0.8           | 0.7960 | 0.3707 | 202.89M |
> > > | Criteo  | [1]    | ebd_dim=log_1.6_{d_i}, lr=0.01, wdcy=1e-6, lambda=1e-3 | 0.8129 | 0.4391 | 357.18M |
> > > | Avazu   | [1]    | ebd_dim=8, lr=0.1, wdcy=1e-8, lambda=1e-5              | 0.7946 | 0.3715 | 206.65M |
> > >
> > > For all models, we use Adagrad optimizer with a batch size of 2048.
> > > Compared to [1], our DPN achieves better performance on Avazu while have similar results on Criteo.
> > > Also, the DPNs have fewer parameters.
> > >
> > > Thanks for your suggestions. We will include these results and give a more detailed analysis in our final version upon the acceptance of this paper.
> > >
> > > [1]Field-wise Learning for Multi-field Categorical Data, NeurIPS 2020

---

### Official Review · Reviewer_AzFU · 2021-11-02

**Correctness:** 3
**Technical Novelty And Significance:** 1
**Empirical Novelty And Significance:** 2
**Recommendation:** 3
**Confidence:** 5

**Main Review:**

Pros:
- The algorithm is deployed in a real-world production system.
- Many variants are proposed and empirically studied.

Cons:
- Not enough novelty. The idea is almost the same as [1] and [2]. Both [1] and [2] conducts feature crossing by using one feature to generate the parameters of a neural network that takes another feature as input. Note that [1] is a recent work that has been deployed in a large-scale real-world e-commerce system as well and is a well-known work among some industry practitioners, especially in China.
- More details about the experimental setup may be needed to assess if the setup is fair. For example, please consider reporting the total number of parameters of each baseline, since sometimes performance can be increased by simply increasing the model's capacity. It seems possible that the so-called state-of-art baselines are not well-tuned.
- The reported +1% improvement in the online A/B test could be meaningless without details about the production systems. For example, +1% improvement in an early-stage business with a weak baseline is not as impressive as +1% improvement in a well-developed business with a strong baseline.
- The writing can be improved.

[1] CAN: https://arxiv.org/abs/2011.05625

[2] A Meta-Learning Perspective on Cold-Start Recommendations for Items. NeurIPS 2017.

**Summary Of The Paper:**

This submission is on modeling feature interactions for CTR prediction. It proposes a framework that follows meta-learning. Specifically, to model the interaction between feature F1 and feature F2, it uses a meta neural-network g(F1) that takes F1 and produces the parameter for another neural network f(F2) that takes F2, i.e., the outcome of feature crossing is f(F2) where f's parameter is g(F1). It outperforms the baselines and is deployed online.

**Summary Of The Review:**

The idea is almost the same as the existing works, especially [1].

---

> ### Author Response · Authors · 2021-11-10
> **Response to Cons.3**
>
> Note that on the real production data, 1% increase in online A/B results is significant and brings about 6 millions dollars lift in the overall advertising income when the model serves on the main trafic. Also, we find our model can perform pretty better on small trafic than a simple baseline. We have stated in the paper our model earned 1‰ auc increase over the highly optimized base model on our ad system, which serves the main traffic of hundreds of millions of active users.
>
> Based on the clarification above, we do not see any major point to justify this reject decision.

---

> ### Author Response · Authors · 2021-11-10
> **Response to Cons.2**
>
> The Reviewer  might have overlooked Table (1)-c. on page 6.  For each baseline in movielen-tag, avazu and criteo datasets, we set the embedding layer with the same hidden size 10, We report the base model parameters in Table1 and Table2. As shown in the following Tables ,  the parameters of  best feature-based DPN are less than the AFN, CIN while field-based DPN almost has less parameters than other baselines. For user behavior modeling, sDPN only increase extra 2% parameters compared to Transformer baseline while have much better performance as shown in Table 5 of our papers.
>
> Table 1: Parameters of different algorithms of feature-based datasets.
>
> | BaseModel         | Movielen-tag | Avazu | Criteo |
> | ----------------- | ------------ | ----- | ------ |
> | FM                | 90k          | 1.5m  | 2.1m   |
> | AFM               | 91k          | 1.6m  | 2.1m   |
> | HOFM              | 2.9m         | 18m   | 107m   |
> | PNN               | 102k         | 195k  | 893k   |
> | CIN               | 153k         | 5.2m  | 4.2m   |
> | AFN               | 242k         | 3.3m  | 7.8m   |
> | CrossNet          | 510          | 3.3k  | 6.6k   |
> | CrossMix          | 47k          | 184k  | 318k   |
> | DNN               | 101k         | 126k  | 449k   |
> | Feature-based DPN | 400k         | 509k  | 2.1m   |
> | Field-based DPN   | 87k          | 219k  | 246k   |
>
> Table 2: Parameters of different algorithms of user behaviors modelings
>
> | BaseModel   | Amazon-Electronic |
> | ----------- | ----------------- |
> | DIN         | 86.9k             |
> | DIEN        | 332.2k            |
> | Transformer | 1.22m             |
> | KFAtt       | 2.17m             |
> | DIN+Heter   | 206.7k            |
> | DIEN+Heter  | 452.1k            |
> | Trans+Heter | 1.46m             |
> | Trans+Homo  | 1.23m             |
> | sDPN        | 1.24m             |

---

> ### Author Response · Authors · 2021-11-10
> **Response to Cons.1 ( con'd)**
>
> ## The difference to NLBA[2] and LWA[2]:
>
> 1. NLBA and LWA mainly focus on the cold-start problem on Twitter recommendation. DPN focuses on the interaction module for traditional and sequence-based CTR prediction task compared to strong baselines.
> 2. Besides the difference in motivation, LWA and NLBA build on the top of matrix factorization (a shallow linear or non-linear classifer), while DPN builds on the top of pointwise methods, e.g. MLP. LWA and NLBA use users’ history as task-dependent weights and bias which can be seen as an instantiation of sequence-based DPO which only utilized in the final layer.  The NLBA don't produce dynamic weights across users in the hidden layers. However the sDPN generate dynamic weights in both homogeneous dynamic convolution operation and heterogeneous query-behavior interaction module.
>
> The related part in original paper is quoted as follows:
>
> >Our first approach to conditioning predictions on users’ item histories has parallels to latent factor
> models and is appealing due to its simplicity: we learn a linear classifier (for new items) whose
> weights are determined by the user’s history $V_j$. Given the two class-representative embeddings $R^0_j ;R^1_j$ described above, LWA provides the bias (first
> term) and weights (second term) of a linear logistic regression classifier.
>
> >In contrast to LWA, all weights (output and hidden) in NLBA are constant across users, while the
> biases of output and hidden units are adapted per user. One can think of this approach as learning a
> shared pool of hidden units whose activation can be modulated depending on the user (e.g. a unit
> could be entirely shot down for a user with a large negative bias).
>
> > Compared to LWA, NLBA produces a non-linear classifier of the item representations $F(t_i)$ and
> can model complex interactions between classes and also between the classes and the test item. For
> example, interactions allow NLBA to explore a different part of the classifier function space that is not
> accessible to LWA

---

> ### Author Response · Authors · 2021-11-10
> **Response to Cons.1**
>
> The reviewer mistakenly determines the DPN has no novelty at all and thinks that CAN[1], NLBA[2] and LWA[2] share the same idea with DPN. We strongly disagree with the reviewer on this point and would clarify the difference from both motivation, approach and experiments:
>
> ## The difference to CAN[1]:
>
>  ### Motivation:
>
> 1.  CAN: aims to capture feature co-action by assuming there exists an optimal function $f_*(A,B)$, and models the function only in the input stage. And they stress the hypothesis the feature co-occurrence can boost the performance by a “CARTESIAN PRODUCT” model. Thus, CAN only cares about how to build an expressive mutual representation for feature pairs at the INPUT STAGE.
>
> The related part in original paper is quoted as follows:
>
> > In this paper, we stress the importance of feature co-action modeling and argue state-of-the-art methods underestimate the importance of co-action seriously. These methods fail to capture the feature co-action due to the limited expressive power. The importance of capturing feature co-action to augment the input is that it can reduce the difficulty for the model to learning and capture the co-action.
>
> > To this end, we propose feature Co-Action Network (CAN) that can capture the feature co-action at the input stage and utilize
> the mutual and common information of different feature pairs effectively.
>
> 2. Our DPN: does not care about feature co-action. DPN addresses the suboptimal combinatory way to build implicit and explicit feature interaction models in previous works [3-6],  by learning both additive and multiplicative interaction in a single module, which can be applied to any stage.
>
> The related part in original paper is quoted as follows:
>
> > The methods mentioned above either model implicit and explicit feature interactions isolated oradopt a suboptimal way to combine them, which can be inefficient. In this work, we aim to address these problems by introducing a small MLP layer that dynamically generates kernels conditioned by the current instance to capture both implicit and explicit feature interactions. The core idea is
> to first generate context weights and biases from the context stream, and then aggregate them with the input stream adaptively. We formulate a generic function and implement it with an efficient dynamic parameterized operation (DPO).
>
> ### Approach:
>
> 1. CAN: can be formulated as $y=DNN(e_{item}, e_{user}, H(x_{user}, x_{item} ; \theta_{can}) ; \theta_{DNN}) $ and $H $means Co-Action Unit. As shown in their paper, the Co-Action Unit only models the user-item interaction in a specific way where they choose $P_{item}$ as the parameter of $MLP_{can}$. In section 4.2 of their paper, CAN seperate the embedding outputs of $P_{item}$ as weight matrix and bias vector followed by matrix multiplication with $P_{user}$ which can be seen as an instantiation of feature-based DPO. The details of CAN can be formulated as:
>
> >$P_{item} = concatenate({flatten(w^{i}}, b)_{i=0,...,k-1})$
>
> >$|P_{item}| = \sum_{i=1}^{k} |w^i| + |b^i|$
>
> >$h^0=P_{user}$
>
> >$h^i=\sigma(w^{i-1} \otimes h^{i-1} + b^{i-1})$
>
> >$H(P_{user}, P_{item})=h^{k}, i \in 1, 2, ... , k-1$
>
> 2. Our DPN: first give an atomic formulation as $y_i= \frac{1}{C(z)} \sum_{\forall j}(f(x_i ; g_i(z_j ; \theta))$, followed by feature-based, field-based, sequence-based variants. As shown in DPN, the inputs and context don’t specified as User and Item which can be utilized to a broader extent. Despite that, DPN discusses more variants than CAN. In Table2.b and Table3 of our paper, DPN compares the performance of different context e.g. User and Item, which covers the CAN. There is no reason to consider DPN the same as CAN. The variants can be formulated as :
>
> > Feature-based DPO: $ y=(\hat{W}^T z+\hat{b})^T x+(\dot{W}^T z+\dot{b})=z^T\hat{W}x+\dot{W}^Tz+\hat{B}^Tx+\dot{b} $
>
> > Homo Behavior DPO: $ y_i = \frac{1}{C(x)} \sum_{l= \lfloor -\frac{k}{2} \rfloor}^{  \lfloor \frac{k}{2} \rfloor} \sum_{j=0}^{t} g_l(x_j ; \theta)  x_{i+l} $
>
> > Hetero Behavior DPO: $ y= g(\frac{1}{C(z)}\sum_{\forall j\in t} z_j ; \theta)^T x$
>
> ### Experiments:
>
> 1. CAN: conducts both feature-interaction and user behavior modeling experiments, while ablation study is far less than DPN. They didn't conduct the experiments on industrial dataset.
> 2. Our DPN: shows more experiments on feature interaction and user behavior experiments, and even gives the results on industrial dataset, which further confirm the online A/B test results.
>
>
> [1] CAN: https://arxiv.org/abs/2011.05625
>
> [2] A Meta-Learning Perspective on Cold-Start Recommendations for Items. NeurIPS 2017.
>
> [3] Deep & Cross Network for Ad Click Predictions, KDD 2017
>
> [4] DCN V2: Improved Deep & Cross Network and Practical Lessons for Web-scale Learning to Rank Systems, WWW 2021
>
> [5] Product-based Neural Networks for User Response Prediction,  ICDM 2016
>
> [6] AutoInt: Automatic Feature Interaction Learning via Self-Attentive Neural Networks, CIKM 2019

---

> > ### Comment · Reviewer_AzFU · 2021-11-18
> > **The contributions still seem incremental.**
> >
> > # Motivation:
> >
> > ## CAN aims to capture co-action:
> >
> > I think that "co-action" is an unnecessary and misleading term coined by the authors of CAN and do not see significant differences between "co-action" and "feature-crossing".
> >
> > ## CAN only cares ... the INPUT STAGE:
> >
> > This point seems irrelevant, because: (1) CAN can be trivially extended to handle the intermedia stages, and (2) more importantly, many of the existing works on feature crossing also focus mainly on the input stage.
> >
> >
> > # Approach:
> >
> > I agree that the exact implementations are different. However, I believe that both [1] and [2] also fulfill the submission's title "Dynamic Parameterized Network". In this regard, this submission is not the first paper that comes up with a "dynamic parameterized network for CTR prediction", which makes the submission's contribution incremental.
> >
> >
> > # Experiments:
> >
> > ## CAN ... didn't conduct the experiments on industrial dataset. DPN ...:
> >
> > CAN is fully deployed in their systems. They certainly have their results on their industrial dataset. To be honest, reporting results on the industrial dataset provides no extra insight if A/B testing results are already reported.
> >
> > ## About the reported results:
> >
> > I share the same concerns with the other reviewers and suspect that the baselines might not be well-tuned.
> >
> > Moreover, I strongly encourage the authors to provide some showcases that can illustrate why the proposed method works and on what kind of data it excels while the baselines do not.

---

> > > ### Author Response · Authors · 2021-11-19
> > > **Response to the Reviewer AzFU**
> > >
> > > ## Response to the Motivation of CAN:
> > >
> > > 1. In CAN, the co-action is defined as the co-occurence of feature A and B which is treated as a new feature corresponding to the baseline model named as “cartesian production”. As shown in their paper, though, such a model induces problems of huge computation and difficulty to learn low frequency features while beating SOTA combinatorial embedding methods such as PNN, NCF, DeepFM. Overall, they propose that pairwise feature combination implemented by CARTESIAN PRODUCT is better than well-known feature-crossing methods but suffers from huge complexity. Thus, they propose CAN to mitigate the de facto of the CARTESIAN while having competitive performance.  What’s the motivation of CAN If “co-action” is the same as “feature-crossing” ?
> > >
> > > 2. We have stressed that CAN is only employed on the input stage without any other extensive experiments in the intermediate stages. However, DPN reparameterize all the neurons by context in feature-based, field-based and sequence-based methods. Our papers propose a generic DPO formulation and provide more ablation studies about how to build weight-generate and feature-aggregation operations in DPO which covers not only the stages where we aim to deploy but also specific implements corresponding to the dataset. We strongly disagree with the statement [2].  The methods of feature-crossing contain a stacked paradigm and parallel paradigm. The first contains IPNN, OPNN. The latter contains DeepFM, XDeepFM, DCN, DCN v2.
> > > We don’t deny the contribution of CAN. The contribution of DPN aims to broaden the stacked and parallel paradigm by introducing explicit and implicit interaction modules into a module together while CAN still follows a stacked paradigm.
> > >
> > > ## Response to Approach:
> > >
> > > We have stated the difference in [1] and [2] by comparing the formulation and implementation both on traditional ctr prediction and user behavior modeling. But the reviewer still sticks to the semantics of the title of our paper which avoids the contribution of our papers. In [1] and [2], they somewhat fulfill the dynamic attribute by introducing multiplication between pairwise features into a submodule of a big model. But we don’t observe any clear claims about the relationship to dynamic neural networks nor citation to the  dynamic neural networks. Please elaborate the evidence of how [1] and [2] clearly claim the dynamic network as shown in our paper and their improvement over static counterparts.
> > >
> > > ## Response to the Experiment
> > >
> > > 1. There are no offline results presented on the industrial dataset in the CAN paper which is an important proof to verify the reliability of the proposed method.The point that results on the industrial dataset provides no extra insight if the A/B testing results reported is nonsense. Just as the reviewer mentioned before, bare online A/B test results are not convincing enough which can be achieved easily with low baselines. It is common to have inconsistent results between offline experiments and online A/B test in the real-world recommender systems due to the complexity of the industrial data. Therefore, consistent performance on both offline industrial experiments and online A/B tests can serve as strong evidence for the effectiveness of the methods. If the reviewer really knows the recommender systems in the real-world production, it is unbelievable to make this statement.
> > >
> > > 2. We have clearly stated the experimental settings in our paper. Please also refer to the response to Reviewer pP6H.
> > >
> > > 3. We have indeed included showcases to show the benefit of our paper on warming up infrequent user embeddings, which is clearly reported in Table 3 and Table 5. The related part in the original paper is quoted as follows:
> > > > We found they share similar results for most experiments while get best performance when we set context as zt  (i.e. use the tag information of context embeddings as context inputs). This handcraft best results mainly originate from the expertise knowledge of MovieLens dataset and recommendation system. Meanwhile, it reveals a nice property of our methods: the intrinsic decoupling attribute can be more separably modeled.
> > >
> > > Enhancing the interaction of non-user features with global embedding reduces the dependence on the sufficiency of user features, and improves the performance on infrequent user embeddings accordingly.
> > > Also, we conducted analysis of relations to the mixture of experts, multiplicative interaction and dynamic convolution in our paper for better understanding the intuitions of our paper. Please refer to Appendix H on page 22 in the revised version.
> > >
> > >
> > > [1] CAN: https://arxiv.org/abs/2011.05625
> > >
> > > [2] A Meta-Learning Perspective on Cold-Start Recommendations for Items. NeurIPS 2017.

---

### Note · Authors · 2024-05-01
**Submission Withdrawn by the Authors**

I have read and agree with the venue's withdrawal policy on behalf of myself and my co-authors.

---

### Decision · Program_Chairs · 2022-01-20

**Decision:**

Reject

**Comment:**

Although scores are somewhat mixed, even ignoring the most negative review the overall score would still be somewhat below the acceptance threshold.

The authors and reviewers had a robust discussion, mostly about the novelty, experimental setting, and the significance of the results. Although the discussion ultimately did not reach a consensus, I think there are valid points on both sides. E.g. I somewhat disagree with the reviewer that the paper is too application-focused for ICLR, though several other points remain valid. The overall message that the experiments seem not totally convincing was highlighted by multiple reviewers.